

# Investigating processes that control the vertical distribution of aerosol in five subtropical marine stratocumulus regions - A sensitivity study using the climate model NorESM1-M

Lena Frey[1,2], Frida A.-M. Bender[1], and Gunilla Svensson[1]

[1]Department of Meteorology and Bolin Centre for Climate Research, Stockholm University, Stockholm, Sweden
[2]Now at: Institute of Meteorology and Climate Research, Karlsruhe Institute of Technology, Karlsruhe, Germany
**Correspondence:** Lena Frey (lena.frey@kit.edu)

**Abstract.** The vertical distribution of aerosols plays an important role in determining the effective radiative forcing from aerosol-radiation and aerosol-cloud interactions. Here, a number of processes controlling the vertical distribution of aerosol in five subtropical marine stratocumulus regions in the climate model NorESM1-M are investigated, with a focus on the total aerosol extinction. A comparison with satellite lidar data (CALIOP, Cloud-Aerosol Lidar with Orthogonal Polarization) shows

that the model underestimates aerosol extinction throughout the troposphere, especially elevated aerosol layers in the two regions where they are seen in observations. It is found that the shape of the vertical aerosol distribution is largely determined by the aerosol emissions and removal processes in the model, primarily through the injection height, emitted particle size, and wet scavenging. In addition, the representation of vertical transport related to shallow convection and entrainment are found to be important, whereas alterations in aerosol optical properties and cloud microphysics parameterizations have smaller effects

on the vertical aerosol extinction distribution. However, none of the alterations made are sufficient for reproducing the observed vertical distribution of aerosol extinction, neither in magnitude nor in shape. Interpolating the vertical levels of CALIOP to the corresponding model levels, leads to a better agreement in the boundary layer and highlights the importance of the vertical resolution.

## 1 Introduction

Aerosol interactions with clouds and radiation constitute a major source of uncertainty in estimates of total radiative forcing. Aerosol particles can scatter and absorb solar radiation, causing a local cooling or heating. The altered temperature profile may in turn induce changes in cloud cover, where the so-called semi-direct effect describing dissipation of clouds in response to local heating, is one out of several possible adjustments (Hansen et al., 1997). The resulting radiative forcing, including the cloud

adjustments to the altered temperature profile, is referred to as effective radiative forcing from aerosol-radiation interactions. Aerosols can further modify the cloud albedo since an increase in the number of aerosol particles leads to more numerous





and smaller cloud droplets for a cloud with a given amount of liquid water content. This enhancement in cloud reflectivity is known as the cloud albedo effect (Twomey, 1977). An increase in cloud droplet number concentration can further lead to suppression of precipitation since the formation of rain droplets is less efficient for a higher number concentration of smaller

cloud droplets, and this rapid adjustment is referred to as the cloud lifetime effect (Albrecht, 1989). The cloud albedo and cloud lifetime effects are summarized as the effective radiative forcing from aerosol-cloud interactions. The overall effect of aerosol-radiation interactions, aerosol-cloud interactions, and the related rapid adjustments is estimated to be negative but with a substantial uncertainty [-0.9 (-1.9 to 0.1) $\mathrm{Wm}^{-2}$] (Myhre et al., 2013). The vertical distribution of aerosols is one important factor for determining the aerosol effect on the radiative budget, both for aerosol interaction with clouds, that requires vertical

co-location, and for aerosol interaction with radiation.

Model intercomparisons and comparisons with observations have shown large disagreement in the vertical distribution of aerosols in general, and absorbing aerosols in particular, with large regional variation (Yu et al., 2010; Koffi et al., 2012, 2016). Model diversity and uncertainty in radiative forcing from aerosol-radiation interaction has been found to be largely referable to the vertical distribution of black carbon (BC), the main absorbing aerosol type (Samset and Myhre, 2011; Samset et al.,

2013). Schwarz et al. (2010, 2013) found that models overestimate BC concentrations over the remote Pacific compared to aircraft observations, whereas the amount of biomass burning aerosols above clouds have been found to be underestimated in models over the southeast Atlantic and often prescribed as too reflective (Peers et al., 2016). This is in agreement with Frey et al. (2017), who found that aerosols above the cloud layer occur in CMIP5 (Coupled Model Intercomparison Project phase 5) models, without reducing the scene albedo.

Highlighting the diversity among climate models, Koffi et al. (2012) compared vertical profiles of aerosol extinction of AeroCom (Aerosol Comparisons between Observations and Models) phase I models with satellite observations, and Koffi et al. (2016) further investigated if models from AeroCom phase II have improved compared to phase I models, focusing on regional and seasonal variability. Although the models were found able to reproduce the general features of the observed aerosol distribution, with a decrease of aerosol extinction from the surface up to 5 km, many models fail to capture the shape

of the aerosol distribution in more detail.

Given its importance for the total aerosol forcing, and its large model diversity and poor agreement with observations, it is important to further investigate which processes are important for determining the vertical distribution of aerosols in global models, and how a better agreement with observations can be reached. Kipling et al. (2016, 2013) accordingly investigated various factors affecting the vertical aerosol distribution in two models (HadGEM3-UKCA and ECHAM5-HAM2), pointing

at the importance of removal processes, which is also supported by the findings from Vignati et al. (2010) who found a large sensitivity of BC lifetime to wet scavenging in a chemical transport model. Studying biomass burning aerosols in particular, Peers et al. (2016) rather point at injection height and vertical transport as the main reasons for discrepancies between their chemical transport model and satellite observations. In the present study, we add to the generalisability of these previous results, by testing the sensitivity to several processes which can control the vertical distribution of aerosol in another climate

model, NorESM1-M. The sensitivity experiments performed are classified into five categories, following Kipling et al. (2016): emissions, transport, microphysics, deposition and aerosol optical properties. Although some of the sensitivity experiments





target specific aerosol types, we focus the evaluation on total aerosol extinction, and number concentration, without discriminating between absorbing and reflecting aerosols, to give a full description of the vertical aerosol distribution in the model, and to facilitate a comparison with observational estimates of total extinction.

While the analysis by Kipling et al. (2016) is on global scale, we focus here on regional scale, and investigate five subtropical marine stratocumulus regions, defined by Klein and Hartmann (1993). The radiative properties of the clouds in these regions, and their potential alteration by aerosol-influence, remain a key challenge in climate models (Bony and Dufresne, 2005; Medeiros et al., 2008; Qu et al., 2014; Bender et al., 2016). Further, both absorbing and reflecting aerosols (BC, organics and dust) located above the cloud layer have been identified in observations (Waquet et al., 2013; Winker et al., 2013; Chand

et al., 2008; Devasthale and Thomas, 2011) of these regions, that display a variety of aerosol signatures in terms of types and column burdens. To evaluate the model performance against observations, we use the 5 km aerosol profile product of CALIOP (Cloud-Aerosol Lidar with Orthogonal Polarization) version 4.10. A description of the satellite data retrievals can be found in Sect. 2, while a description of the climate model NorESM1-M and the performed model simulations is provided in Sect. 3. The results and further discussion are presented in Sect. 4 and 5, respectively. We summarize the most important processes that

control the vertical aerosol distribution in the climate model NorESM1-M in the given regions in Sect. 6, and thereby give a guidance to evaluating and improving this and other state-of-the-art climate models.

## 2   Satellite retrievals and data processing

CALIOP is on board the CALIPSO (Cloud-Aerosol Lidar and Infrared Pathfinder Satellite Observations) satellite as part of the A-train constellation. The satellite was launched in year 2006, and we used data for the time period 2007 to 2016. We used

the Level 2, 5 km aerosol profile product, version 4.10, of CALIOP lidar data, which has shown a better agreement with the aerosol optical depth (AOD) from observations, compared to the previous CALIOP version (Kim et al., 2018).

CALIOP measures the backscattered radiation at two wavelengths using a lidar and detector and derives the aerosol extinction, with an algorithm including iterative adjustment of the lidar ratio, i.e. the ratio between extinction cross section and 180° backscatter cross section. While e.g. Yu et al. (2010) and Koffi et al. (2012, 2016) use the CALIOP aerosol layer product, which

can lead to an underestimation of aerosols near the surface, we use here the 5 km aerosol profile product (cf. Winker et al., 2013), which provides profiles of the total aerosol extinction coefficient. A detailed product and data processing algorithm description can be found in Winker et al. (2009) and Kim et al. (2018). Only wavelength 532 nm is considered here, as these measurements have a better signal-to-noise ratio than those at wavelength 1064 nm (Yu et al., 2010). Due to the higher detection sensitivity for aerosols in the night (Winker et al., 2009, 2010), we further use only night-time data, cf. Yu et al. (2010)

and Koffi et al. (2012). We apply several additional data screening criteria, following Tackett et al. (2018). The cloud-aerosol discrimination (CAD) score distinguishes between clouds and aerosols, with a negative CAD score representing aerosol and a positive value representing cloud. We use here a CAD score range between -80 and -20, for a higher confidence in identifying aerosol (Liu et al., 2009, 2018). We also examine the quality of the extinction retrieval, represented by the extinction QC filter, which stores information about the initial and final state of the lidar ratio at each layer. We use only cases where the initial





lidar ratio remains unchanged during the iterative solution process, referred to as an unconstrained retrieval (QC flag =0) or use constrained retrievals (QC=1), where the initial lidar ratio was adjusted during the retrieval process by using measurements of a layer two-way transmittance, both with a higher confidence in the algorithm solution. Furthermore, we reject retrievals with a high extinction uncertainty of $99.9 \ \mathrm{km}^{-1}$, avoiding thereby high-biases in aerosol extinction.

Our analysis focuses on five regions of low marine stratocumulus clouds, following Klein and Hartmann (1993); Australian

(25-35°S, 95-105°E), Californian (20-30° N, 120-130°W), Canarian (15-25°N, 25-35°W), Namibian (10-20°S, 0-10°E) and Peruvian (10-20°S, 80-90°W). The CALIPSO satellite overpasses the equator twice per day. The temporal resolution of the lidar is 5 seconds and the snapshots for each given satellite overpass are aggregated to a uniform 2° latitude by 5° longitude grid with a vertical resolution of 60 m, so that each provided data file within each region contains multiple aerosol profiles. We average all profiles in the latitude- and longitude range of each region to obtain a daily mean profile; a minimum number

of ten profiles at each vertical layer is thereby required to avoid high-biases in aerosol extinction in the upper troposphere. In addition, to allow for a better comparison with the coarser model resolution of 26 vertical layers, we interpolate the daily mean lidar profiles to the altitudes corresponding to the model levels. The daily mean profiles are further averaged over the whole 10-year period to obtain a climatological annual mean.

## 3   Model and model simulations

### 3.1   Model NorESM1-M

The atmospheric part of the climate model NorESM1-M (Kirkevåg et al., 2013) is based on the Community Atmosphere Model version 4 (CAM4; Gent et al., 2011) and coupled to the aerosol module CAM4-Oslo. The horizontal resolution is 1.9° for latitudes and 2.5° for longitudes and the vertical is resolved with 26 levels from 1000 hPa up to 0.1 hPa using hybrid-sigma-pressure coordinates. Here an AMIP-configuration of the uncoupled model version, with a prescribed sea surface temperature

and sea ice climatology, was used.

Aerosol types represented in the model are mineral dust, sea salt, organic matter (OM), black carbon (BC) and sulfate. Mineral dust emissions are prescribed and inserted at the surface while sea salt emissions are prognostic and wind-driven. Anthropogenic aerosol emissions of sulfate, primary OM and BC from fossil fuel and biofuel combustion, and biomass burning are in the default model configuration based on the IPCC AR5 data set (Lamarque et al., 2010). Biofuel and fossil fuel emissions

are injected at the surface whereas biomass burning emissions are distributed over the lowest eight model levels. Emission heights follow the recommendations by Dentener et al. (2006).

Nucleation, condensation, coagulation and aqueous chemistry processes are represented, and the emitted particles are tagged with one of these production mechanisms. The aerosol scheme in NorESM1-M is a sophisticated aerosol module, where all aerosol particles can be internally mixed, i.e. absorbing particles can become reflecting and active as cloud condensation nuclei

(CCN). All aerosol types are mainly reflecting, except BC which is prescribed as fully absorbing. In terms of aerosol-cloud interactions, both the cloud albedo and cloud lifetime effects are parameterized. The cloud droplet effective radius ($r_{eff}$) is dependent on the cloud droplet number concentration ($N_d$), which is dependent on the aerosol number concentration and



vertical velocity through supersaturation, based on the parameterization by Abdul-Razzak and Ghan (2000). Suppression of precipitation with increased aerosol number concentration (lifetime effect) is triggered by two thresholds in the autoconversion

scheme, a critical radius from which cloud droplets are converted to rain droplets and a maximum precipitation rate. Mean aerosol size distributions and optical properties are calculated a posteriori using look-up tables. The aerosol mass concentration is production-tagged and also calculated offline.

All aerosol particles can be removed by dry and wet deposition. For convective clouds an in-plume approach is used, i.e. the convective cloud cover is calculated explicitly and aerosols in convective clouds can be removed directly by wet scavenging,

cf. Zhang and McFarlane (1995). The boundary layer scheme is based on Holtslag and Boville (1993) using an updated representation of the boundary layer height, cf. Vogelezang and Holtslag (1996).

Further information of the model can be found in Kirkevåg et al. (2013).

### 3.2 Model setup and sensitivity experiments

All model simulations are run in an AMIP-type configuration with a prescribed sea surface temperature and sea ice clima-

tology at representative of preindustrial conditions. Only anthropogenic aerosol emissions are increased to present-day level, corresponding to the year 2000. Following Kipling et al. (2016) we use an on/off approach for analyzing the sensitivity to several processes, and in other cases use an observationally motivated parameter range. Sensitivity simulations with changes of processes influencing the vertical distribution of aerosol were performed and a control simulation serves as a reference. This experiment setup isolates changes in aerosol distribution, driven by the selected processes. We note here, that changes in the

sensitivity experiments are applied globally, so that effects in the focus regions may also be driven by changes on the larger scale. The single-process approach taken here differs from methods of statistical sampling of a broad parameter space to identify key drivers of uncertainty, which has been demonstrated by e.g. (Lee et al., 2011, 2012, 2013) to be useful for investigating sources of uncertainty in model representation of CCN.

Our methods target specific processes relevant for the vertical aerosol distribution, and in combination with the limited

geographical distribution and dynamical similarity of the focus regions, we can isolate factors for which there are physical reasons to expect an effect on the vertical distribution in the given areas. The on/off approach (cf. Kipling et al., 2016), rather than mimicking realistic variations, highlights the importance of basic physical processes and their representation in the model for the vertical distribution of aerosol. We note that the results of the performed sensitivity are limited to the individual parameters and ranges chosen, and that potential effects of interaction between processes and parameters can not be uncovered,

cf. Lee et al. (2011).

All model simulations were run for a simulation time of 10 years, following a 1-year spin-up period. A summary of all experiments can be found in Table 1 and a more detailed description of all experiments, divided into the categories of emissions, deposition, vertical transport, microphysics, and aerosol optical properties, following Kipling et al. (2016), is presented in the following.





### 3.2.1 Emissions

Magnitude, altitude and type of emissions, or anthropogenic aerosol sources, directly affect the distribution of aerosol. In this category of sensitivity experiments we vary emission data set, emission height as well as emitted particle size. For all cases except the altered emission data set, the total emitted aerosol mass is kept constant.

For the default model configuration, the IPCC AR5 emission data set (Lamarque et al., 2010) was used. Fire emissions in the default data set are based on the Global Fire Emissions Database (GFED) version 2, and aviation emissions are not included. An additional aerosol emission data set, combining emissions from the Evaluating the Climate and Air Quality Impacts of Short-Lived Pollutants (ECLIPSE) project (Stohl et al., 2015) version 3 and updated fire emissions from the GFED version 3.1 (van der Werf et al., 2010) as well as aviation emissions, representative of the year 2010 is implemented in the experiment Aero2010. As the altered emission data set represents a later emission year, differences between the default and alternative emission data set can encompass interannual variability besides differences in the data set construction.

In NorESM1-M, biomass burning aerosols (consisting of BC and OM) are emitted at eight model levels. The sensitivity to the emission height of biomass burning aerosols is tested here using four experiments with varying emission height. For the first experiment all biomass burning emissions were inserted at the lowest model emission level (Aero2000_surface_inj), and in the second one all biomass burning emissions were inserted above the cloud layer at the highest predefined emission level at approximately 510 hPa (Aero2000_high_inj). The third experiment inserts biomass burning aerosols uniformly over all emission levels (Aero2000_uniform_inj). Finally, all biomass burning aerosols were injected at the lowest three model levels, which are within the boundary layer in these regions (Aero2000_PBL_inj).

The size of primary emitted particles can influence the vertical distribution, through changes in removal and transport processes. Due to the large variability in the control simulation (standard deviation up to 76%), we test the sensitivity to particle size by increasing and decreasing the radii of primary emitted particles by as much as $\pm$ 50% in two experiments (Aero2000_aero_small_50 and Aero200_aero_large_50, respectively).

### 3.2.2 Deposition

Deposition constitutes the main aerosol sink, and is hence also of direct relevance to the aerosol distribution in the model. All aerosol types are affected by wet and dry deposition in the model, and here an on/off approach was used to study the sensitivity to these two main removal processes (Aero2000_nowetdep and Aero2000_nodrydep). Dry deposition takes the particle size into account and has an additional gravitational settling for coarse particles. Wet deposition represents in-cloud and below-cloud scavenging, whose impact was broken down into two separated experiments, allowing only below-cloud (Aero2000_noscav_incloud) and only in-cloud scavenging (Aero2000_noscav_belowcloud), respectively. In-cloud scavenging refers to nucleation and impaction processes, through which aerosols can enter cloud droplets whereas below-cloud scavenging refers to aerosol removal through liquid precipitation.



### 3.2.3 Vertical transport

For given sources and sinks, transport can further affect the vertical aerosol distribution in the model and vertical transport of aerosols is primarily controlled by convection. To test the sensitivity of the aerosol extinction profile to convective transport, the original convection scheme was replaced with a modified version which assumes full mixing of aerosols between up- and downdrafts in convective clouds (Aero2000_convmix), see Seland et al. (2008). Furthermore, in one experiment shallow convection parameterization was switched off completely (Aero2000_noshallowconv), affecting not only the convective transport of aerosols, but also of heat, moisture and momentum. As the model resolution is too coarse to resolve convection, it is an extreme scenario to turn off the shallow convection scheme, but it emphasizes the importance of shallow convective transport for the vertical distribution of aerosols. Aerosols are also vertically displaced by entrainment of dry air into the moist cloud layer. The sensitivity to entrainment was studied, again using an on/off approach (Aero2000_noentrain) and disabling entrainment for convective clouds.

### 3.2.4 Cloud microphysics

Activation of aerosols to form cloud droplets, and conversion of cloud droplets to rain drops are microphysical processes that can affect the vertical distribution and properties of aerosols. In this category of experiments, we target microphysical parameterizations in the model.

We first vary the efficiency of the auto-conversion, i.e. the transformation of cloud water into rain water, which in turn controls removal of aerosol particles through wet deposition. In addition to the extreme scenario to switch off auto-conversion in warm clouds (Aero2000_noautoconv), two more parameters that control the auto-conversion rate in NorESM1-M were changed; the critical droplet radius for the onset of autoconversion was decreased from the default value of 14 to 5 $\mu m$ (Aero2000_rcrit_autoconv_5) and the maximum precipitation rate for the termination of autoconversion was decreased from the default of 5.0 to 1.0 mm $day^{-1}$ (Aero2000_precip_autoconv_1).

The activation of cloud droplets depends on the vertical velocity on cloud-scale. NorESM1-M uses a characteristic subgrid vertical velocity, which is parameterized through the turbulent diffusion coefficient and a constant characteristic mixing length (cf. Morrison and Gettelman, 2008). Due to a high variability of the control simulation, the default value of 10 m$s^{-1}$, based on Morrison and Gettelman (2008), was increased to an extreme value of 30 m$s^{-1}$ in the sensitivity experiment Aero2000_omegamin_30.

### 3.2.5 Aerosol optical properties

To address the fact that aerosols above clouds tend to be insufficiently absorbing in models (Peers et al., 2016), we also alter the aerosol optical properties in the model. Peers et al. (2016) found that climate models with a refractive index for BC of 0.71 show a better agreement with satellite observations compared to models with a refractive index of 0.44. Here, BC is prescribed as fully absorbing with a default imaginary part of the refractive index of 1.00, but to test the sensitivity to this optical property





we decreased it to 0.44 (Aero2000_BCrefrac_044) and 0.71 (Aero2000_BCrefrac_071), making the pure BC in the model more reflecting.

### 3.3 Model output and post-processing

To evaluate the effects of the sensitivity experiments on the vertical aerosol distribution, monthly model output was used, and profiles of total aerosol extinction coefficient and aerosol number concentration compared. The mean aerosol profiles were obtained by averaging all grid points in each of the focus regions at each vertical model level (cf. Koffi et al., 2012, 2016). As shown by Koffi et al. (2012), collocating the model grid to match CALIOP coordinates causes only little variation to averaged regional aerosol profiles, indicating that the regional coverage by CALIOP is sufficient for the averaging method used here.

In addition, the aerosol column burden, i.e. a mass measure of aerosols, as well as the cloud droplet number concentration where clouds are present, are investigated. The monthly model output is averaged over the 10 year simulation period to obtain a climatological mean.

## 4 Results

### 4.1 Regional characteristics

The focus regions are similar in regard to dynamical regime, but differ in their aerosol signature (e.g., Frey et al., 2017). These subtropical marine stratocumulus regions are located in the subsiding branch of the Hadley cell, and the capping inversion limits the vertical cloud extent.

Figure 1 shows the column burden of the five aerosol types represented in the model relative to the total column burden for the control simulation. In all regions, the largest contribution to the total column burden comes from dust and sea salt aerosols

in agreement with Textor et al. (2006), but in the Namibian and Peruvian regions biomass burning aerosols (including both BC and OM) account for almost 50% of the total aerosol burden. The Canarian region located downwind of the Sahara desert is dust-dominated and the Californian region has a high contribution of sulfate aerosols compared to other regions.

### 4.2 Observed vertical aerosol extinction distribution

Figure 2 shows the vertical distribution of the total aerosol extinction coefficient retrieved from CALIOP in comparison with

the model control simulation for the five focus regions. The vertical resolution of CALIOP data is higher than the coarse model resolution, and CALIOP vertical levels were interpolated to the equivalent model levels to facilitate comparison (see Sect. 2). Figure 2 shows both the original and the coarser-resolution versions of the CALIOP profiles. To indicate the variability of the model control simulation, we use a $\pm 1$ standard deviation range of the monthly model output, which is referred to as the uncertainty range in the subsequent analysis of the sensitivity experiments. The variability is greatest in the dust-dominated

Canarian region, which is also the region where the magnitude of the extinction coefficient is highest for both observations and model output.





The original CALIOP distribution of aerosol extinction shows an increase in magnitude in the boundary layer and then a decrease throughout the troposphere, except in the Namibian and Canarian regions, where local maxima in aerosol extinction occur above the boundary layer. The interpolated CALIOP distribution does not show the maximum in the boundary layer seen in the original CALIOP distribution and shows instead a decrease from the surface throughout the boundary layer. With few

minor exceptions, the model underestimates the magnitude of the aerosol extinction for all regions and levels, and in addition the shape of the distribution in the vertical differs between model and observations. If compared to the original CALIOP distribution, the model has difficulties to represent the distinct observed maximum in aerosol extinction in the boundary layer, in agreement with the findings of Koffi et al. (2012). If compared to the interpolated CALIOP distribution, the model distribution

shows a better agreement in the boundary layer with a decrease in extinction from the surface throughout the boundary layer. This indicates that the model resolution is too coarse to resolve relevant processes in the boundary layer. However, the elevated aerosol layers in the Canarian and Namibian regions, seen both in the original and the interpolated CALIOP distributions, are underestimated and not well represented in the model. This indicates that resolution is not the limiting factor for representing the above-cloud aerosol layer.

### 4.3 Sensitivity experiments

The large regional variations, and discrepancies between models and observations motivate the wide ranges used in the sensitivity tests, the results of which are shown in the following. For clarity, only a selected subset of experiments are visualized for each of the five experiment categories.

#### 4.3.1 Emissions

The choice of an alternative aerosol emission data set (Aero2010) yields an increase in aerosol extinction and aerosol number concentration, mainly in the lower troposphere in the biomass burning regions (see Fig. 3), but only in the Peruvian region the increase in aerosol number concentration falls outside the uncertainty range of the control simulation ($\pm 1$ standard deviation, based on monthly means for 10 years). A decrease in both aerosol extinction and number occurs in the other regions. The ECLIPSE emission data set of the year 2010 compared to the model's default IPCC AR5 data set of the year 2000 shows a

higher total aerosol optical depth (AOD) and absorption aerosol optical depth (AAOD) in the biomass burning regions (see Fig. 4).

The variation in injection height of biomass burning aerosols affects, as expected, mainly the two biomass burning regions, particularly the Namibian region. Inserting all biomass burning aerosols higher up in the free troposphere (Aero2000_high_inj), leads to a higher aerosol number concentration and extinction in the upper troposphere and a decrease in the lower troposphere

(Fig. 3). Shifting the insertion to the surface (Aero2000_surface_inj), leads to a reduction in aerosol number and extinction throughout the troposphere (not shown). Choosing a uniform insertion over all emission levels (Aero2000_uniform_inj), leads to a similar distribution as in the control simulation, and only in the Canarian and Namibian regions an increase in aerosol number and extinction occurs above the boundary layer (not shown). Emitting all biomass burning aerosols in the boundary layer (Aero2000_PBL_inj) yields a significant increase in extinction throughout this layer and also above in the Namibian





region and leads to an improved distribution compared to the observations. Nevertheless, the observed distribution with a local maximum of extinction in the boundary layer can not be reproduced by the model.

All experiments, except the experiment with the use of an alternative emission data set (Aero2010), are mass conservative, i.e. the same total aerosol mass was emitted. Hence, by changing the size of primary emitted particles both aerosol size and number distribution are affected. Increasing the size (Aero2000_aero_large_50) shifts the distribution to larger but fewer particles and subsequently yields a decrease in aerosol extinction, with the strongest response in the Canarian region (not shown). Decreasing the size of all particles (Aero2000_aero_small_50) leads to the opposite effect with an increase in aerosol number concentration, especially in the Namibian and Peruvian regions and an increased aerosol extinction, up to eight times higher than for the control simulation in the Canarian region (see Fig. 3). The increase in number concentration is more similar across regions, and hence can not explain the stronger increase in extinction in the Canarian region. As a consequence of the change in size distribution, the aerosol composition changes as well, as an effect of changes in the aerosol lifecycle (e.g. removal processes). A comparison of the regional aerosol burden characteristic of the control experiment (Fig. 1) and the sensitivity experiments Aero2000_aero_small_50 and Aero2000_aero_large_50 (Fig. 5) shows an increase in the dust column burden in all regions subsequently of the decrease in size, since the smaller dust particles are less affected by gravitational settling. This increase in the dust column burden yields in turn an enhanced absorption and therefore higher extinction in the Canarian region. Furthermore, an increase of the column burden of biomass burning aerosols occurs in the Namibian and Peruvian regions. Similarly, increasing the size of particles shifts the composition towards a higher sea salt and lower dust burden in all regions (see Fig. 5).

In the Canarian, Peruvian and Namibian regions a change in the shape of the vertical distribution can be noticed in response to the decrease in size with a more pronounced maximum in aerosol extinction in the boundary layer.

### 4.3.2 Deposition

Disabling one of the removal processes leads in all cases to an increase of aerosol number concentration (see Fig. 6), but the effect is greatest when wet deposition is cut off (Aero2000_nowetdep). Changes in aerosol extinction and number due to disabling dry deposition are small and within the given uncertainty range of the control simulation (Aero2000_nodrydep). All aerosol species are affected by dry and wet deposition, but dry deposition is primarily important for particles in the coarse mode, like dust and sea salt.

The dominant removal process of aerosols in the model is wet deposition, and the in-cloud wet scavenging accounts for most of the total wet deposition (Aero2000_noscav_incloud). Hence, the experiment with disabled wet deposition and in-cloud scavenging give similar effects on the vertical aerosol distribution (see Fig. 6), while only little effect was found for disabling below-cloud scavenging (Aero2000_noscav_belowcloud, not shown). Altering the deposition influences not only the amount of aerosols, but also the shape of the vertical distribution. While the control simulation shows a steady decrease of aerosol extinction in the boundary layer, disabling wet deposition and in-cloud scavenging leads to an increase with a maximum in the boundary layer, similar to the observed distribution.





In the Californian region, the aerosol number concentration shows a small increase (within uncertainty) compared to the control simulation, and in the Canarian region even a decrease in number in the boundary layer is seen with disabled wet

deposition, while the aerosol extinction shows a strong increase (see Fig. 6). This can be explained by a shift in aerosol composition resulting from alteration of the deposition sinks. Figure 7 shows the relative column burden contribution of the different aerosol types in the focus regions. The aerosol composition is shifted towards a higher burden of sulfate aerosol in all regions in response to the cut off wet removal. Furthermore, in the Australian, Namibian and Peruvian region the dust burden increases while a decrease occurs in the Californian and Canarian regions. This shift in composition affects the extinction more

than the changes in number concentration. Switching off dry deposition gives no significant shift in aerosol composition.

### 4.3.3   Vertical transport

The modified convective scheme (Aero2000_convmix) results in a small decrease in aerosol number concentration and extinction within the uncertainty throughout the troposphere in the focus regions (see Fig. 8).

Disabling shallow convection, aerosols remain closer to the surface leading to a strong increase in aerosol number and

extinction in the boundary layer compared to the control simulation (Aero2000_noshallowconv). Resulting changes in aerosol extinction are thereby beyond the ±1 standard deviation uncertainty range of the control simulation, in all regions. The shape of the vertical distribution is altered and shows a strong increase in the boundary layer and a decrease above the boundary layer (see Fig. 8).

A similar response was found from switching off entrainment for convective clouds (Aero2000_noentrain, see Fig. 8). The

aerosol number and extinction decreases in the boundary layer, especially in the biomass burning regions and increases in the upper troposphere. The importance of this process on the cloud droplet number is shown in Fig. 9. The cloud droplet number concentration shows a decrease when entrainment is disabled, especially in the Namibian region.

### 4.3.4   Microphysics

The effect of varying several autoconversion-related parameters is shown in Fig. 10. The chosen processes on the microphysical

scale have only a weak impact on aerosol extinction and number concentration with changes within uncertainties of the control simulation (not shown here are Aero2000_rcrit_autoconv_5 and Aero2000_precip_autoconv_1). Only the extreme scenario with no autoconversion in warm clouds (Aero2000_noautoconv), leads to an increase in aerosol extinction that reaches beyond the given uncertainty range in the lower troposphere in all regions. The shape of the vertical distribution is not notably affected by the changes in this subset of microphysical processes (see Fig. 10).

Focusing on cloud properties, the altered autoconversion efficiency has more impact. Figure 11 shows the vertical distribution of cloud droplet number concentration and effective radius for the control simulation and sensitivity experiments. The strongest response occurs from a changed subgrid vertical velocity (Aero2000_omegamin_30), with a significant increase in cloud droplet number concentration and a decrease in effective radius. Changes due to autoconversion being switched off (Aero2000_noautoconv) are within the ±1 standard deviation uncertainty.





### 4.3.5 Aerosol optical properties

Decreasing the default value of the imaginary part of the refractive index from 1.0 to a value of 0.44 (Aero2000_BCrefrac_044) and 0.71 (Aero2000_BCrefrac_071), makes BC more reflecting. This does not affect the aerosol number concentration, and Fig. 12 shows the single scattering albedo (SSA, i.e. the fraction of extinction that is due to scattering) together with the total extinction, to illustrate the effects of the change in BC optical properties. The SSA shows in both experiments an increase, i.e.

a higher fraction of reflection, as expected. The changes in aerosol extinction are however small and within the uncertainty of the control experiment.

## 5  Discussion

Discrepancies between the control simulation and CALIOP satellite data were found in all focus regions, with regard to the total aerosol extinction as well as shape of the vertical distribution. In particular, the model underestimates the absolute values

of aerosol extinction, showing a steady decrease from the surface while observations indicate a maximum in the boundary layer. An adaptation of the CALIOP vertical resolution to the equivalent model resolution gives a better agreement. The maximum in the boundary layer is not captured with a coarser, model-like, vertical resolution for CALIOP. This emphasizes the importance of the vertical resolution to resolve mixing and transport processes in the lower troposphere. However, the model also underestimates aerosol extinction of elevated aerosol layers seen in two regions in the observations even if compared to

the adapted CALIOP resolution.

It is also worth noting that while the observations are taken from the period 2007-2016, the emissions used in the model simulations (except in the Aero2010 experiment) are for the year 2000, and that year-to-year variability in aerosol emissions may contribute to discrepancies between observed and modelled vertical profiles.

The sensitivity experiments performed suggest that the alterations that have the largest impact on the aerosol vertical profiles

are found in the categories emissions, deposition and vertical transport, whereas changes in the categories microphysics and aerosol optical properties have less effect. However, none of the chosen alterations of parameters and processes affecting the vertical distribution of aerosol extinction in the model are sufficient to reproduce the observed distribution. For instance, the emission height of biomass burning aerosols directly influences the aerosol vertical profile. Inserting these absorbing aerosols above or within the boundary layer, leads to an increased aerosol extinction above the boundary layer, as expected. Biomass

burning aerosol injection at the surface only, or uniformly in height has less effect on the vertical profile, in agreement with (Kipling et al., 2016), and none of the altered emission height simulations reproduces the local maximum in the boundary layer produced by the original-resolution satellite data.

The choice of the aerosol emission inventory was also found to be important for determining the magnitude of total vertically integrated aerosol extinction, in agreement with the findings of Kirkevåg et al. (2013). By choosing aerosol emissions for the

375 year 2010 a higher extinction and subsequently a higher AOD was produced, especially in biomass burning-dominated areas. Considering the small interannual variability in biomass burning aerosol emissions from the main burning regions, found by Giglio et al. (2010), the differences between the two emission data sets are more likely related to differences in resolution and





data collection than to interannual variability. As discussed in Giglio et al. (2010) and van der Werf et al. (2010), emissions in GFED3 have increased compared to GFED2 due to an improved mapping approach of burned areas using MODIS and a

380 higher resolution of 0.5 °compared to GFED2 with 1 °resolution. Previous studies have also pointed at the importance of the spatial (Possner et al., 2016) and temporal resolution (Dentener et al., 2006) of aerosol emissions.

In terms of the vertical aerosol distribution, the updated emission data set leads only to a small change, within the uncertainty range of the control simulation. Kipling et al. (2013) similarly showed that using GFED3 instead of GFED2 biomass-burning emissions leads only to a moderate improvement of the vertical BC distribution compared to observations without statistical

significance.

Another important factor which can control the vertical distribution of aerosol is the size of emitted aerosol particles. The performed sensitivity experiments are mass conservative, except the experiment with an alternative emission data set, meaning that changes in emission particle sizes lead to a shift in the entire size- and number distribution. Here we find, that the shape of the vertical distribution in the model is highly sensitive to the size of emitted particles. Decreasing the size results in more

numerous smaller particles and produces a maximum in aerosol extinction in the boundary layer in the Canarian, Namibian and Peruvian regions. This is not only an effect of changes in aerosol number concentration and size distribution, but also of the resulting shift in aerosol composition produced by the model in response to the change in size distribution.

Large responses were also seen in the sensitivity experiments focusing on removal processes, particularly for the cases of

395 altered wet deposition. Dry deposition mainly affects larger particles and cutting this sink off leads to a small decrease in extinction, throughout the vertical column. Wet deposition on the other hand affects in particular smaller particles, and is the major removal process for aerosol particles in the model. Cutting off this removal pathway leads to a large increase in extinction, and a modified shape of the vertical distribution. In-cloud scavenging contributes more than below-cloud scavenging to the total wet deposition, and hence turning off in-cloud scavenging has similar effects as turning off wet-deposition completely, while

turning off below-cloud scavenging has little effect, in agreement with (Kipling et al., 2016; Vignati et al., 2010). Hence, the representation of wet deposition is important for the vertical aerosol distribution in the model, in agreement with the findings of Vignati et al. (2010), Croft et al. (2009, 2010) and Kipling et al. (2013). Changes in the removal processes also affect the aerosol composition in the model. Inhibited wet deposition increases the amount of sulfate, BC and OM, as this is the main removal process for these aerosol types, but decreases the relative amount of dust, which is less affected by this removal process. The

smaller portion of wet deposition that is due to below-cloud scavenging also affects composition, but is less efficient for Aitken or accumulation mode particles, a size range where e.g. BC is found.

Kipling et al. (2013) discussed the coupling between wet scavenging and convective transport and its importance for the representation of the vertical aerosol distribution, comparing HadGEM-UKCA with ECAHM5-HAM2, and with observations. The in-plume approach, with wet scavenging directly linked to the convective scheme, implemented in NorESM1-M is in line

with the recommendations in Kipling et al. (2013).





Disabling either of the convective schemes, shallow or deep convection, does not switch off convective transport of aerosols completely, i.e. switching off shallow convection still allows deep convection and vice versa. However, the complete inhibition of this convective scheme largely affects the aerosol distribution. Without the shallow convection scheme, i.e. allowing only

415 deep convection, the shape of the vertical distribution changes with a more pronounced increase close to the surface. Particles remain closer to the surface as they can not be lifted higher, leading to an increase in aerosol number concentration and extinction especially in the boundary layer. Hence, shallow convection in the model is essential for transporting aerosols to the middle troposphere in the focus regions, consistent with Kipling et al. (2016) who showed that vertical transport of aerosol is dominated by convective processes on unresolved scales on the global scale. Hoyle et al. (2011) highlighted further the impor-

420 tance of the parameterization of convective processes for tracers with a short lifetime. Another important transport process for aerosols is entrainment, and cutting off this mixing for convective clouds results in a decrease in extinction in the boundary layer and an increase in the upper troposphere in the biomass burning regions. However, the entrainment particularly controls the amount of aerosols above the boundary layer and is crucial for the formation of cloud droplets via provision of CCN.

Microphysical processes, though linked to wet removal processes, have less impact on the vertical aerosol distribution. Altering the process of autoconversion results only in small changes in aerosol number and extinction and only the extreme scenario of disabling autoconversion completely in warm clouds, leads to a significant increase in aerosol number and extinction in the boundary layer. Focusing on changes in cloud droplet number concentration, however, autoconversion and the subgrid vertical velocity are more important processes in the model. This is in agreement with previous studies pointing at the importance of

the autoconversion parameterization for aerosol indirect effects (e.g. Rotstayn and Liu, 2005; Golaz et al., 2011) and the representation of the cloud lifetime effect in models (Michibata and Takemura, 2015), as well as at the importance of the subgrid variability of the vertical velocity when estimating aerosol indirect effects (Golaz et al., 2011).

Finally, turning to optical properties, our results indicate that they have little impact on the vertical aerosol profile. Peers et al. (2016) point at the amount of aerosol above clouds as being underestimated in amount, but too reflective in climate

models. They found an improved representation of model output compared to satellite observations for climate models with an imaginary part of the refractive index of 0.71 compared to models with a lower value of 0.44. The refractive index was thereby defined at a wavelength of 0.55 µm. NorESM1-M has a high default refractive index for pure BC with a value of 1.0, so that BC is prescribed as full absorbing for the entire visible spectrum.

In contrast to other models, however, BC can be internally mixed and coated, thereby becoming more reflective. A decrease

of the refractive index causes almost no change in the extinction coefficient. The single scattering albedo (SSA) on the other hand is increased as expected. Hence, while Peers et al. (2016) found that aerosol above clouds in climate models underestimate absorption, primarily due to the properties of BC, our results indicate that for NorESM1-M it is rather the aerosol amount than the optical properties of pure BC that determines the aerosol extinction above clouds.





## 6 Conclusions

In this study the sensitivity of the climate model NorESM1-M to changes in processes affecting the vertical aerosol distribution was studied, focusing on five regions of subtropical marine stratocumulus clouds.

To evaluate the model performance, a control simulation was compared with satellite-borne lidar observations from CALIOP. The magnitude of aerosol extinction is underestimated in the model, and displays a differently shaped vertical distribution. Discrepancies are of similar magnitude to those found for other models (see Koffi et al., 2016) and the main difference in

shape is the lack of local maximum in aerosol extinction in the boundary layer, which is also a common feature among many previously investigated models. The model also underestimates aerosol extinction of elevated aerosol layers above the boundary layer, seen in observations in two of the studied regions.

None of the alterations made here were sufficient for reproducing the observed aerosol extinction, but a better agreement between observations and model in terms of the shape of distribution in the boundary layer was found by interpolating the

455 vertical resolution of observations to the model levels. This highlights the importance of the vertical model resolution to capture aerosol processes especially in the boundary layer. Observed local extinction maxima above the boundary layer appear in observations with both original and reduced vertical resolution, indicating that the model resolution does not restrict here the representation of aerosol layers above clouds.

Among the categories in which sensitivity experiments are performed, the largest impact on the vertical distribution of

460 aerosol extinction is found to result from alterations to emissions, deposition and vertical transport, and less from microphysics and aerosol optical properties. In this sense, the presented results show a general agreement with (Kipling et al., 2016) who conducted similar sensitivity experiments using a different model and focusing on the global mean. In particular, for our model the parameters and processes found to have the greatest effect on the shape of the vertical aerosol distribution in the dynamical regime studied, are the altitude of emissions and size of emitted particles, as well as the representation of shallow convection,

entrainment and wet scavenging.

By emitting all biomass burning aerosol at the highest injection level or within the boundary layer in the model, an increase in aerosol extinction above the boundary layer can be produced, but is still underestimated compared to the local maxima seen in observations in two regions. Hereby, emitting aerosol at higher altitude or within the boundary layer are the most efficient way of increasing extinction above cloud level, which highlights the importance of mixing processes in the boundary layer.

The shallow convection scheme is also important for transporting aerosols up from the boundary layer and by disabling shallow convection, the aerosol extinction increases in the boundary layer. However, the resulting profile has a much too strong increase in aerosol extinction towards the surface, compared to observations, and does not indicate an improved agreement with the observed shape, compared to the control experiment.

Disabling of in-cloud scavenging leads to a maximum in aerosol extinction in the boundary layer, in qualitative agreement

with observations. Similar changes in vertical aerosol distribution are seen when the size of emitted particles is reduced. This qualitative improvement of the modelled aerosol profile suggests that wet scavenging might be too efficient in the model and that the emission size distribution may be shifted towards too large particles.



With a focus on a specific dynamic regime, our sensitivity experiments indicate which processes have the greatest potential to influence the vertical distribution of aerosol in NorESM1-M, finding a general agreement with previous studies based on other models. Our results hereby support and give guidance to further improvement of the representation of aerosol distribution, and thereby aerosol-cloud interactions in this and other state-of-the-art climate models.

*Data availability.* The CALIPSO data are available online at https://www-calipso.larc.nasa.gov/ (NASA, 2017). Data of model simulations can be provided upon request by the corresponding author.

*Code and data availability.* Data produced with model simulations using the Norwegian climate model NorESM1-M and code for data analysis is available from the corresponding author upon request.

*Author contributions.* LF and FB developed the concept of the paper. LF designed and performed all model simulations and data analysis and wrote the manuscript. FB and GS contributed to the design of experiments, interpretation of the results as well as writing of the manuscript.

*Competing interests.* The authors declare that they have no conflict of interest.

*Acknowledgements.* Model simulations, data post-processing and data analysis of model output were performed on resources provided by the Swedish National Infrastructure for Computing (SNIC) at the National Supercomputer Centre at Linköping University (NSC). We would like to thank the Evaluating the Climate and Air Quality Impacts of Short-Lived Pollutants (ECLIPSE) Project no. 282688 and Global Fire Emissions Database (GFED) for providing the aerosol emission data sets and the NASA Langley Research Center Atmospheric Science Data Center for providing CALIOP data. Further, we would like to acknowledge the Vetenskapsrådet, project no. 2018-04274. We would like to thank Alf Kirkevåg and Øyvind Seland from the Norwegian Meteorological Institute in Oslo for providing model-specific information. Special thanks also to Abhay Devasthale who gave advice about the CALIOP aerosol profile product.





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

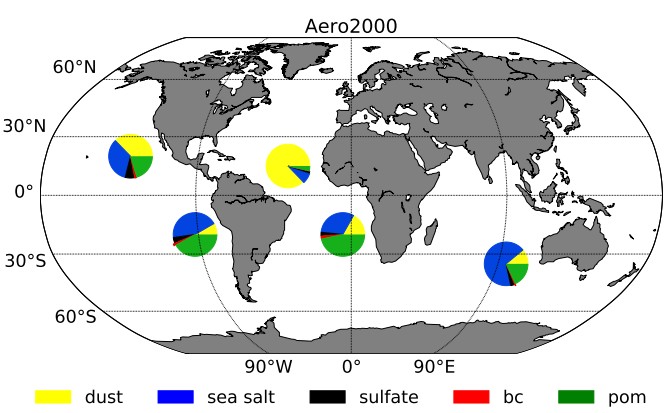

**Figure 1.** The relative columnar burden contribution of each aerosol type to the total column burden in the control simulation for the Australian, Californian, Canarian, Namibian and Peruvian regions.

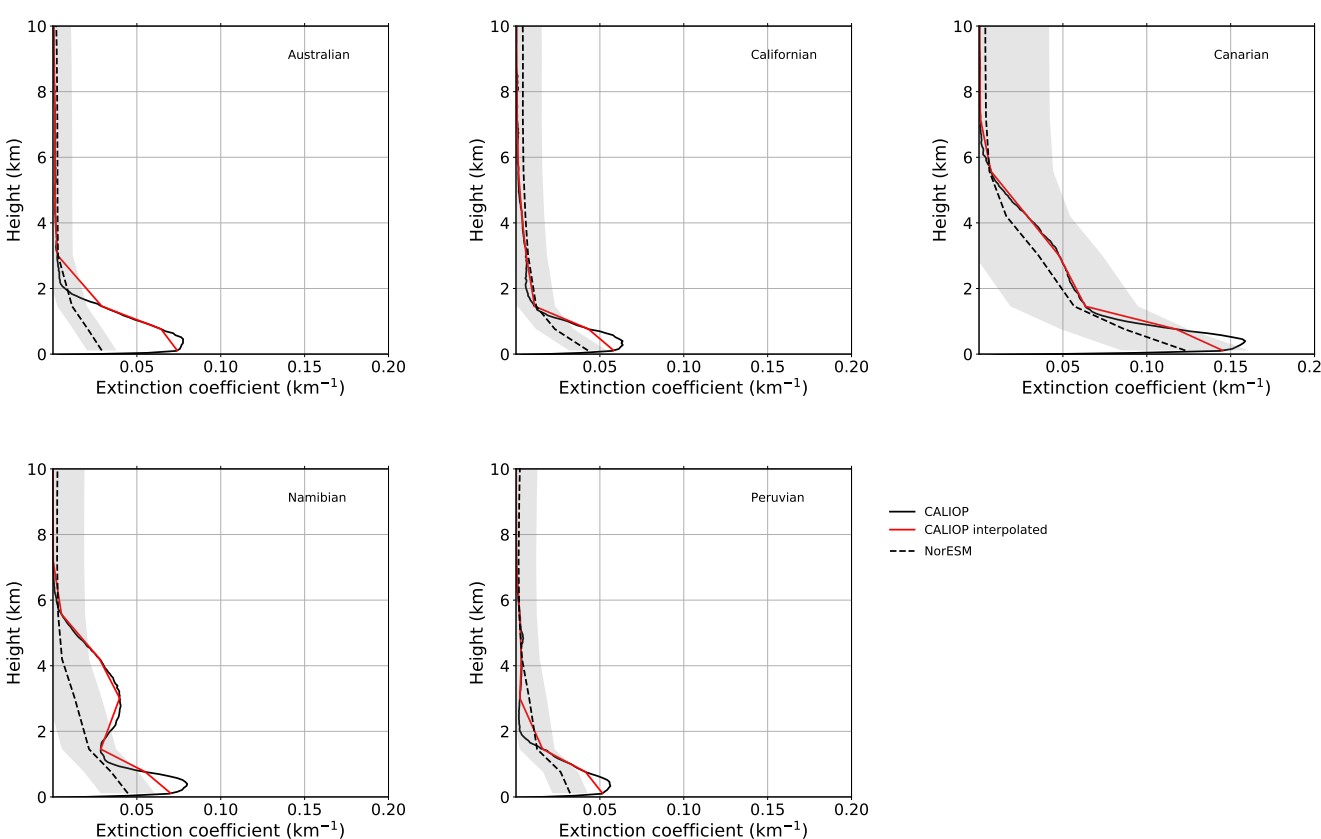

**Figure 2.** Vertical distribution of total aerosol extinction coefficient in $km^{-1}$ for CALIOP data from 2007 to 2016 for the Australian, Californian, Canarian, Namibian and Peruvian region (solid black line). The CALIOP vertical levels were interpolated to the corresponding model levels (solid red line). In addition, the model control simulation averaged over 10 years is shown (dashed line) with the standard deviation (grey shaded area).

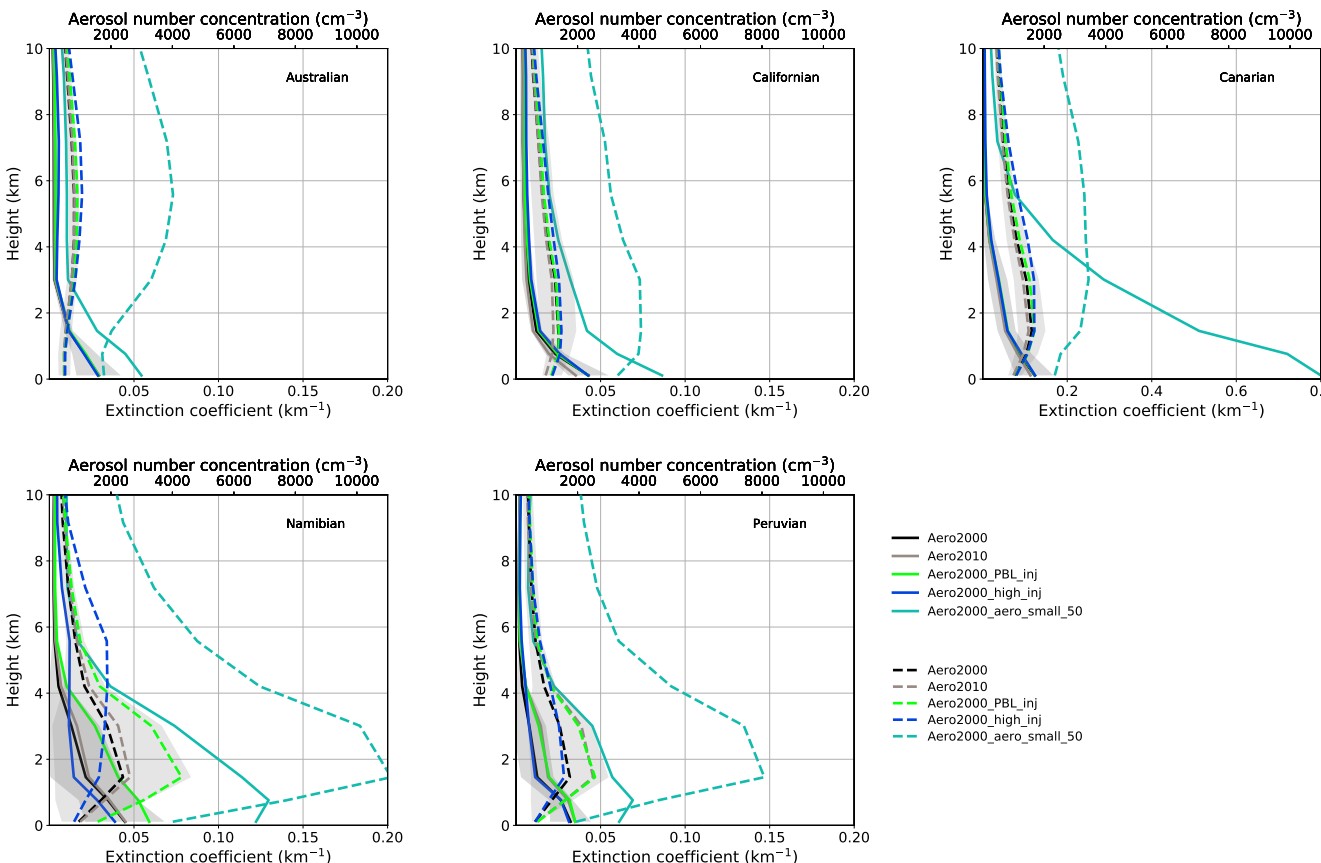

**Figure 3.** Vertical distribution of the aerosol extinction coefficient in $km^{-1}$ (solid line) and aerosol number concentration in $cm^{-3}$ (dashed line) for the Australian, Californian, Canarian, Namibian and Peruvian region for the model control simulation and sensitivity experiments in the category emissions. The standard deviation of the model control simulation is indicated as grey shaded area.





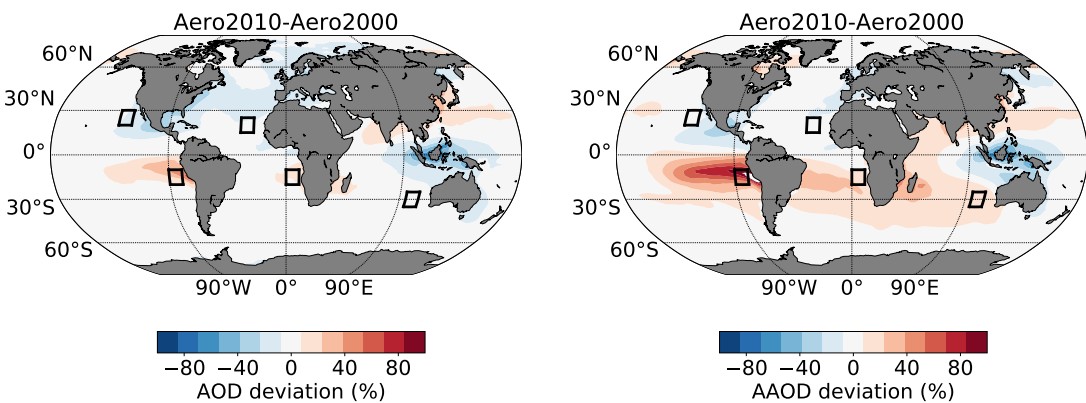

**Figure 4.** Global distribution deviations in aerosol optical depth and absorption aerosol optical depth deviations between the sensitivity simulation Aero2010 and the control simulation Aero2000.



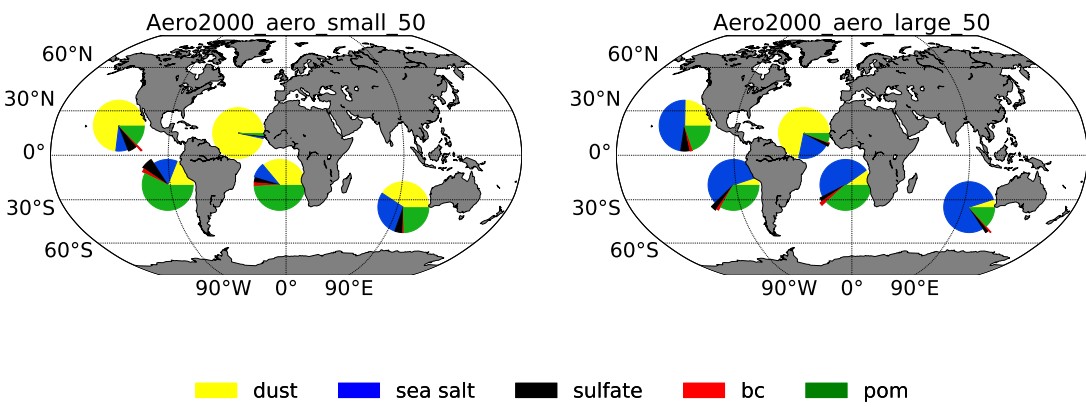

**Figure 5.** The relative columnar burden contribution of each aerosol type to the total column burden in the simulations Aero2000_aero_large_50 and Aero2000_aero_small_50 in the Australian, Californian, Canarian, Namibian and Peruvian region. A shift in composition can be seen compared to the control simulation (see Fig.2).



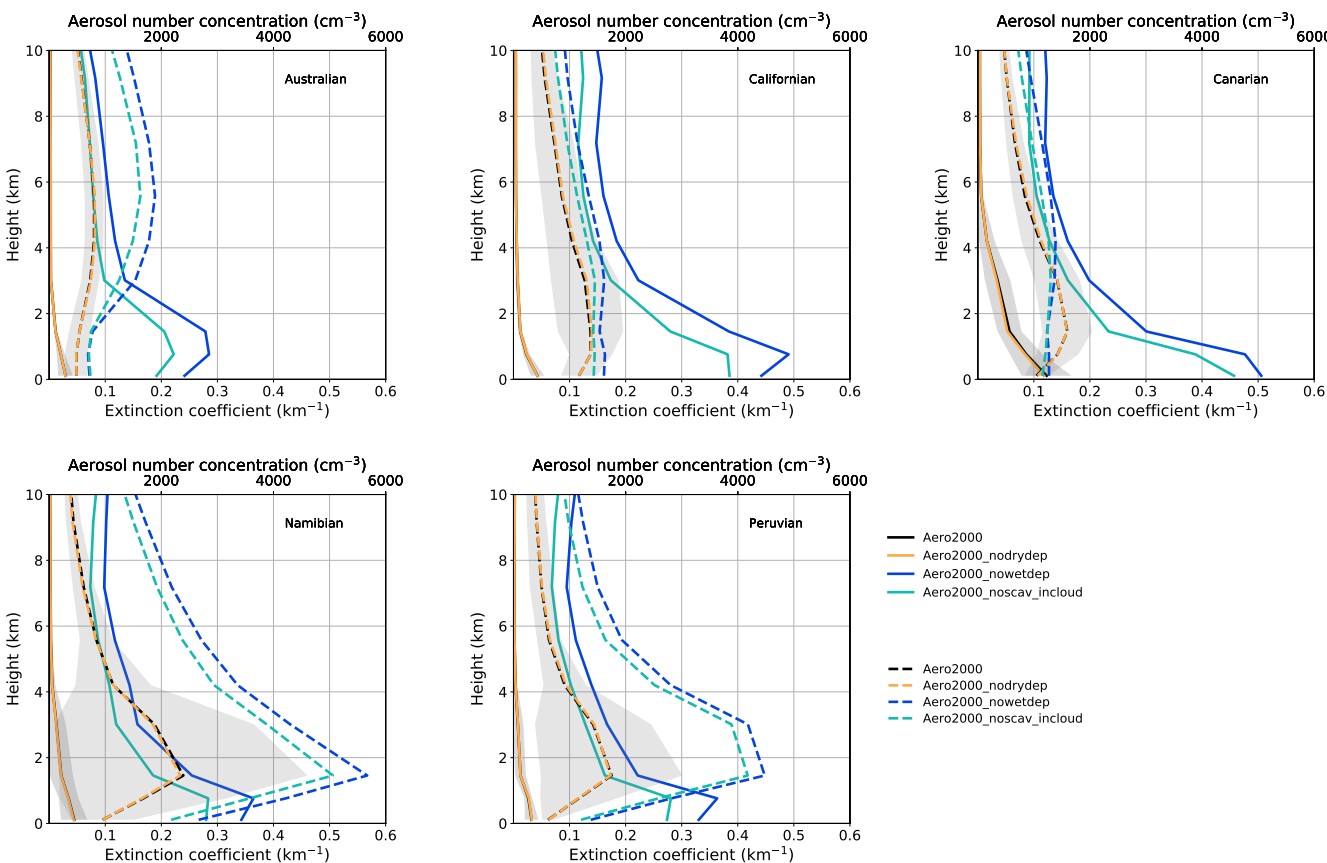

**Figure 6.** Vertical distribution of aerosol extinction coefficient in $km^{-}1$ (solid line) and aerosol number concentration in $cm^{-3}$ (dashed line) for the Australian, Californian, Canarian, Namibian and Peruvian region for the model control simulation and sensitivity experiments in the category deposition. The standard deviation of the model control simulation is indicated as grey shaded area.





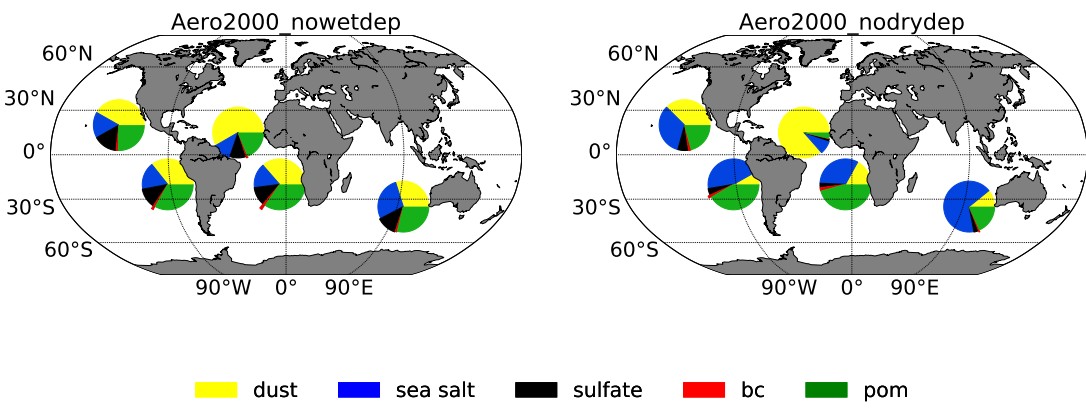

**Figure 7.** The relative columnar burden contribution of each aerosol type to the total column burden in the simulations Aero2000_nodrydep and Aero2000_nowetdep in the Australian, Californian, Canarian, Namibian and Peruvian region.

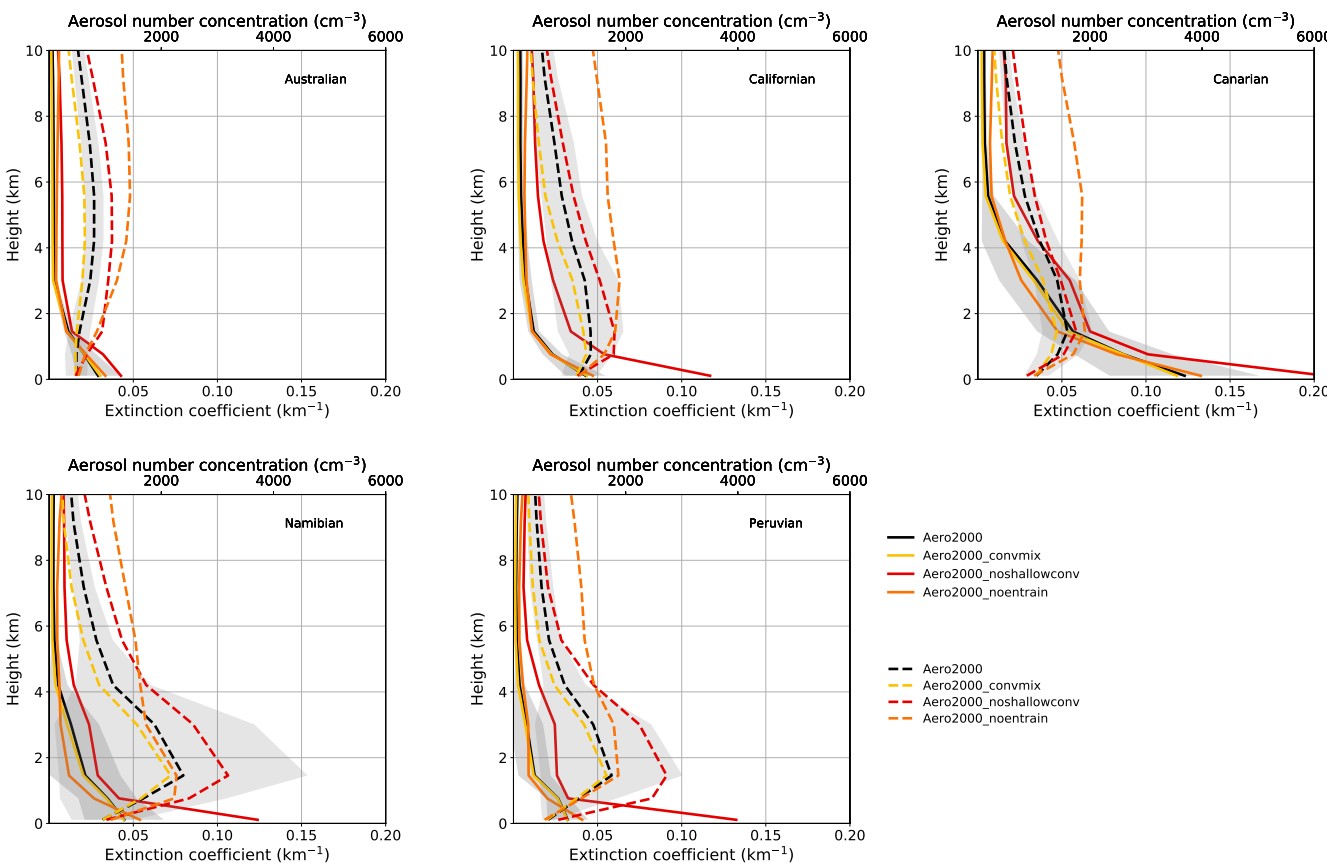

**Figure 8.** Vertical distribution of aerosol extinction coefficient in $km^{-1}$ (solid line) and aerosol number concentration in $cm^{-3}$ (dashed line) for the Australian, Californian, Canarian, Namibian and Peruvian region for the model control simulation and sensitivity experiments in the category transport. The standard deviation of the model control simulation is indicated as grey shaded area.

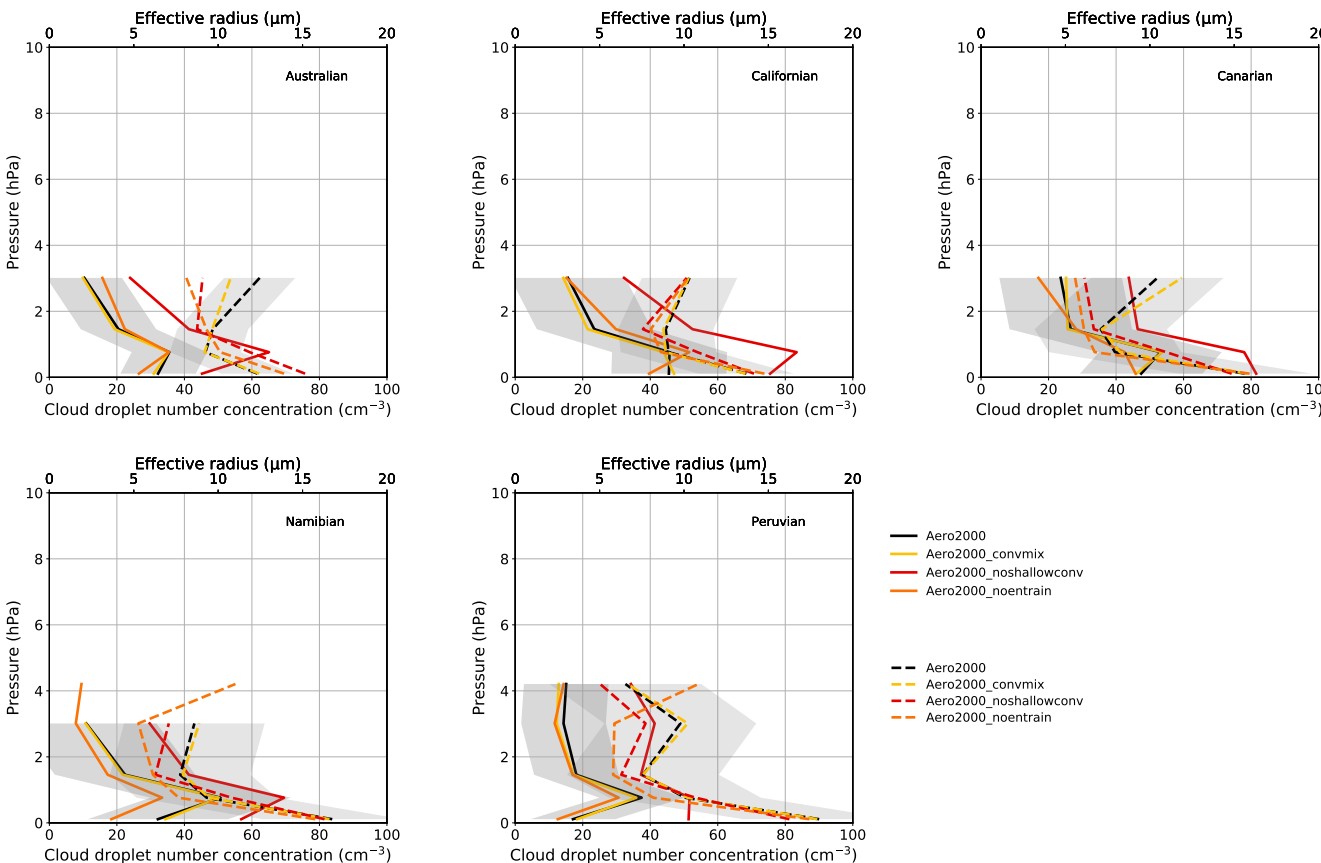

**Figure 9.** Vertical distribution of cloud droplet number concentration (solid line) and effective radius (dashed line) for the Australian, Californian, Canarian, Namibian and Peruvian region for the model control simulation and sensitivity experiments in the category transport. The standard deviation of the model control simulation is indicated as grey shaded area. Both cloud droplet number concentration and effective radius stop at the cloud top.



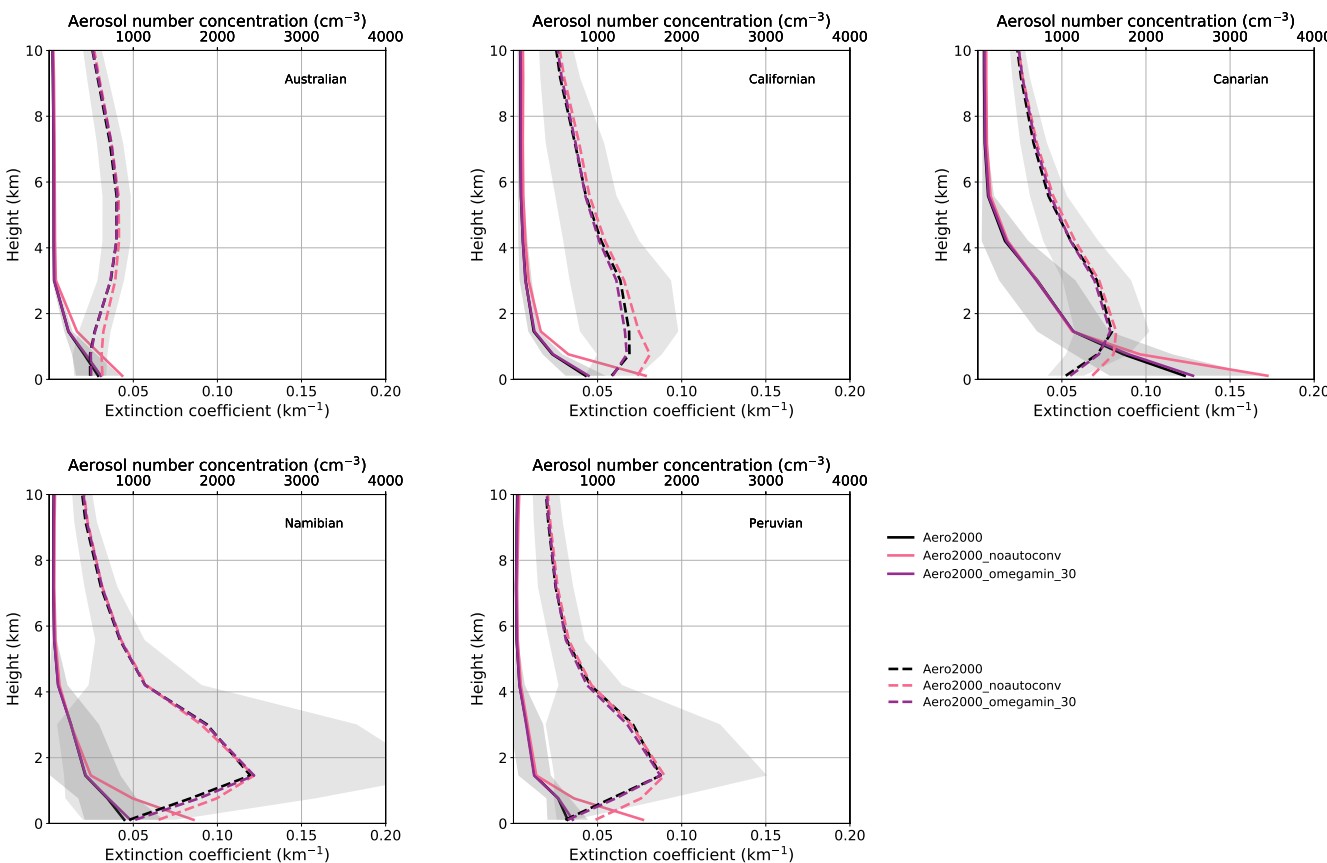

**Figure 10.** Vertical distribution of aerosol extinction coefficient in $km^{-1}$ (solid line) and aerosol number concentration in $cm^{-3}$ (dashed line) for the Australian, Californian, Canarian, Namibian and Peruvian region for the model control simulation and sensitivity experiments in the category microphysics. The standard deviation of the model control simulation is indicated as grey shaded area.



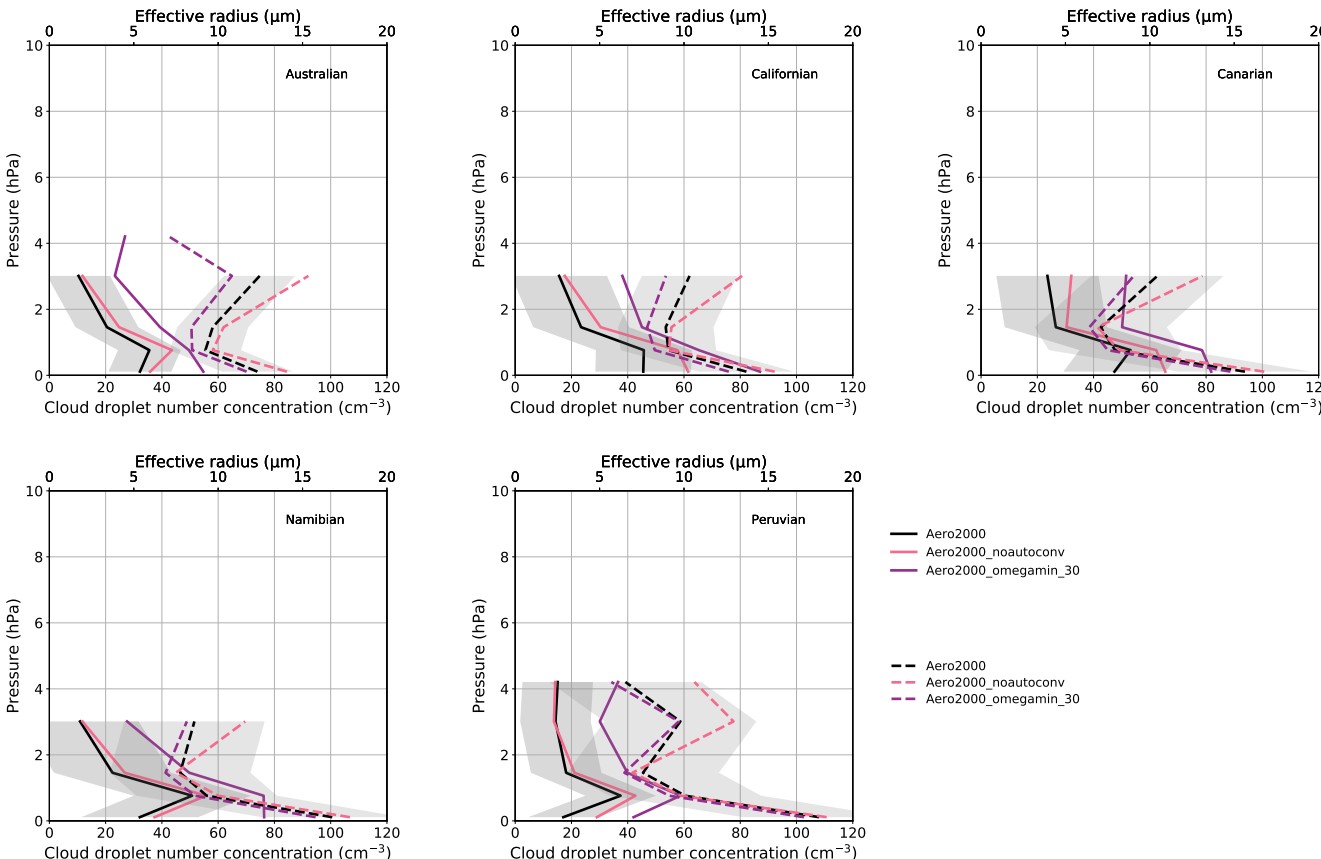

**Figure 11.** Vertical distribution of cloud droplet number concentration (solid line) and effective radius (dashed line) for the Australian, Californian, Canarian, Namibian and Peruvian region for the model control simulation and sensitivity experiments in the category microphysics. The standard deviation of the model control simulation is indicated as grey shaded area. Both cloud droplet number concentration and effective radius stop at the cloud top.

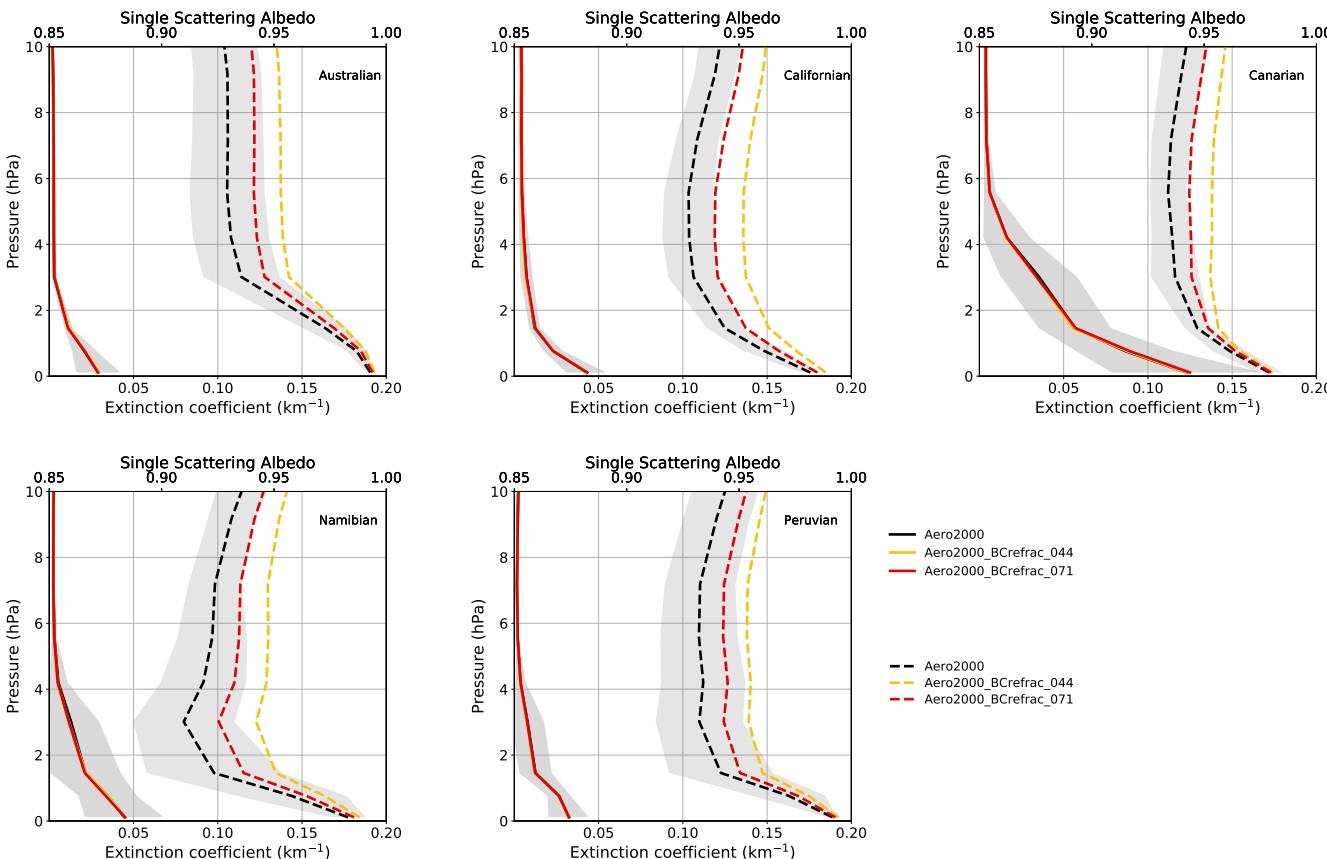

**Figure 12.** Vertical distribution of aerosol extinction coefficient in $km^{-}1$ (solid line) and single scattering albedo SSA (dashed line) for the Australian, Californian, Canarian, Namibian and Peruvian region for the model control simulation and sensitivity experiments in the category aerosol optical properties. The standard deviation of the model control simulation is indicated as grey shaded area.



**Table 1.** Summary and short description of control and sensitivity experiments.

| | Experiment name | Experiment description |
|---|---|---|
| | Aero2000 | Control experiment |
| **Emissions** | Aero2010 | ECLIPSE aerosol emissions from 2010 |
| | Aero2000_surface_inj | BC aerosol emissions inserted at the lowest model emission level |
| | Aero2000_uniform_inj | BC aerosol emissions inserted uniformly in height |
| | Aero2000_high_inj | BC aerosol emissions inserted at the highest model emission level |
| | Aero2000_PBL_inj | BC aerosol emissions inserted at the lowest three model emission levels |
| | Aero2000_aero_small_50 | emitted particle size decreased by 50 % |
| | Aero2000_aero_large_50 | emitted particle size increased by 50 % |
| **Transport** | Aero2000_noshallowconv | no aerosol transport by shallow convection |
| | Aero2000_convmix | improved convective mixing of aerosols |
| | Aero2000_noentrain | no entrainment for convective clouds |
| **Deposition** | Aero2000_nodrydep | no dry deposition |
| | Aero2000_nowetdep | no wet deposition |
| | Aero2000_noscav_belowcloud | no scavenging |
| | Aero2000_noscav_incloud | no scavenging in cloud |
| **Microphysics** | Aero2000_noautoconv | no autoconversion for warm clouds |
| | Aero2000_precip_autoconv_1 | lower maximum precipitation rate for offset of autoconversion |
| | Aero2000_rcrit_autoconv_5 | critical radius of cloud droplets changed to 5μm |
| | Aero2000_omegamin_30 | maximum of sub-grid vertical velocity set to $30\mathrm{ms}^{-1}$ |
| **Properties** | Aero2000_BCrefrac_044 | BC refractive index changed to 0.44 |
| | Aero2000_BCrefrac_071 | BC refractive index changed to 0.71 |