# Peer review of "What are the processes controlling the vertical aerosol distribution in marine stratocumulus regions - A sensitivity study using the climate model NorESM1-M"

_Atmospheric Chemistry and Physics, 2019_

## Referee Comment (RC1) · Anonymous Referee #2 · 8 Dec 2019

General Comments

This article provides an interesting sensitivity study exploring the effect of changes in model parameters and aerosol emissions on aerosol composition and vertical distribution of extinction and number concentration, focussing on the marine stratocumulus regions. It analyses separately the impact of changing parameters one by one in the simulations, and concludes on the relative importance of the processes considered, showing that although some of them like the wet scavenging have a strong impact, none is able to reproduce the CALIOP observations.

[Figure]

I think the analysis could be deepen and the interpretation of the results would gain in being extended. Even if the full chain of processes is very complex to analyses in a climate model, and without providing a full pathway analysis that would need extensive additional work, I think more insight could be gained by crossing the results and trying to interpret them (especially when they are surprising or when there are regional differences).

More direct comparisons with the observations could be provided to asses the effect of parameter changes, and spatial and temporal colocation could increase the robustness of the comparison (although they might not be straightforward to implement). More highlights could be put on answering the question: Could the model possibly represent better the observations if the relevant parameters where adjusted ? Would this set of parameters be realistic? Or are there fundamental discrepancies that cannot be resolved by parameter changes?

I think more simulations could be performed to either better distinguish between processes (convection parameterisation vs. aerosol transport by convection for instance) or investigate other key properties of the model, like its vertical resolution which could be essential in representing the low-level aerosol distribution. More details could also be provided on the model setup, on the characteristics of the parameterisations, and the choices made for the sensitivity study.Please refer to my specific comments hereafter for more details.

The paper is well written overall (some English editing is needed here and there, cf. my technical comments) and is organised in a straightforward way. Although I have numerous comments providing ways for clarifications and improvements, I believe this paper is a good contribution to the literature and I am sure its revised form will de publishable in Atmospheric Chemistry and Physics.

Specific comments

L11-13: is that really resolution that matters here or rather proper colocation? Similarly,

not sure about the relevance of interpolation (cf comments hereafter).

L 80: "... underestimation of aerosols near the surface" any reference supporting this statment?

l99: To be fully consistent, model data should be also extracted along CALIPSO over-passes, at the times of the overpasses, before being averaged. Although daily mean works rather well in areas where there is no strong diurnal cycle in aerosols, proper spatial and temporal colocation (of the model data onto CALIOP measurements) reduces errors (cf. e.g. Schutgens et al., 2017). It may not be easily doable to extract profiles along CALIPSO track from the model, but discussions of sampling errors could be included.

L101-102: Is is really interpolation that is used here? As the CALIPSO data are on a finer vertical grid than the model, it would be better to average all the CALIOP data points located inside one model gridbox than to interpolate between two CALIOP levels to get the value at the central point of the gridbox.

L 110-111: could you please show the location of the vertical model levels at least in one of your plots (e.g. adding markers figure 1) or / and give the spacing between levels in the low to mid troposphere?

L 115: "the lowest eight levels" corresponding to what altitude (on average)?

L 121: be more specific: the cloud albedo and cloud lifetime effects are not directly parameterised, but the microphysics parametrisation takes aerosol into accounts and hence aims to represent them.

L 125: why is there a maximum precipitation rate? It seem odd if you do not specify here (as line 205) "before the autoconversion is switched off".

L 127: what means "production-tagged"?

L 129: "for convective clouds an in-plume approach is used i.e. the convective cloud

cover is calculated explicitly": explain a bit more. What do you mean by "in-plume approach" how is calculated the convective cloud cover? How is it then passed to the large-scale? As convective clouds are parametrised, their cloud cover is surely not fully explicit. As there is no aerosol in the Zhang and McFarlane (1995) scheme, could you be more precise and, if they have been added in a more recent version, cite the relevant literature?

L 134: be more specific on the characteristics of the run, and/or give reference for AMIP setup.

L 159-165: Justify the choice of this emission dataset. Are they more realistic for the simulation period? Why not using realistic monthly emissions for the period of simulations as a control? And then either a different dataset, or a multiplicative factor on emissions for the sensitivity experiment?

L 183: are aerosols also liberated by evaporation of cloud droplets and raindrops? If yes, you could mention it in paragraph 3.1.

l 189: the original convection scheme should be described a bit more (here, or maybe rather section 3.1). Do you mean only deep convection here? What mean the full mixing of aerosols? the aerosol population is the same in updraughts and downdraughts? How about the impact of lateral entrainment then? By "the original scheme" do you mean Zhang and McFarlane (1995) which has no aerosol at all?

L190-196: - What happens in the model when shallow convection is turned off? Is it picked-up by the deep convection scheme? Or by the large-scale as it tends to be when all convection parameterisation if turned off? In any case, turning off shallow convection will not prevent the vertical transport needed to balance surface SW heating and atmospheric LW cooling. - Then, why not turning off only aerosol transport from convection parameterisation (looking at both shallow and deep convection separately)? That would give much clearer results on what is done by the parametrisation in term of aerosols, without having any direct impact on the dynamics, clouds, etc.

l200: what schemes are used?More description of the original scheme is needed (here or in section 3.1) before discussing its perturbations.

L 204-206: You could include the equations for autoconversion (and possibly accretion). This threshold on the radius has been introduced in models historically, partly to compensate for the lack of below-cloud evaporation, but there should not be any threshold as the processes are continuous. The threshold in precipitation is even more arbitrary as cloud droplet should continue to form raindrops no matter how much precipitation there is already (although accretion will then become much more significant than autoconversion, meaning in practice autoconversion might be of little or no effect). Unless the way the equations are written makes it unphysical, I would suggest trying to remove the two thresholds.

L 208: what is the "characteristic subgrid vertical velocity"? What will be the effect of changing it? Explain so that the reader can understand what the chosen values mean.

L 209: "high variability" in what sense?

L 220: is the monthly output obtained from online averaging over the month?

L 222-225: following my previous comment, is that also true for temporal sampling? Schutgens et al. (2016, 2017) suggest the opposite.

L241: again, is it really an interpolation? Averaging would be better.

L 249: what is the average BL height in these regions?

L 256: Indeed model resolution is too coarse (and probably also in the free troposphere up to ∼ 5.5 km). From figure 2, I guess AOD is also underestimated by the model? An interesting additional sensitivity experiment could be to refine the vertical grid in the BL and up to about 5.5 km.

Section 4.3: why not include CALIOP extinction profiles in the figures? This would be useful to compare not only sensitivity experiments with the control but also with

the observations and see when they perform better than the control. Indeed, one big question is whether or not changes in model configuration can lead to results closer to the observations, so a more direct comparison is needed in the figure and in the analysis.

L 283: "Hence, by changing the size of ..." rephrase to make it clearer, e.g. "Hence, changing the size of emitted particles also leads to changes in emitted aerosol numbers"

l 289-290: "As a consequence . . .." I do not understand. Something is not right in the way this sentence in constructed. Please clarify / rephrase.

L 303: The differences from tunring off dry deposition are actually almost non-existent, indicating that the dry deposition plays very little role (if any) in your simulations. Although dry deposition will affect mostly the biggest aerosols, I am a bit surprised that the impact is so small. How big is the impact on the total aerosol burden? Using CAMS, Wu et al (2018) show significant impact of the dry deposition scheme on BC burden (cf. for instance their figure4), and I suppose this could also be the case for dust (Johnson et al, 2012 ). Could you discuss that a bit? Do you think the dry deposition could be underestimated in your control simulation?

L 312: again, it would be helpful to plot the observed extinction profiles on the same figure.

l319-320: can you explain and justify this statement? How do you know the composition changes affects extinction more than the number concentration?

L320: again, I am surprised by the total lack of sensitivity to dry deposition.

Section 4.3.3: more careful description and analysis is needed: -l 326-327: No, there is no decrease in aerosol number above the BL according to fig 8. In all regions and at all heights, there is an increase in both extinction and number concentration. Can you interpret that? It might be related to other changes in the simulation without shallow

convection; turning off only aerosol transport by shallow convection would make the interpretation easier. - L 329: do you mean you switched off entrainment completely in shallow and deep convective clouds (no lateral or below cloud entrainment of aerosol, momentum, environmental air, etc)? Turning off only the transport of aerosols by convection, but keeping entrainment unchanged otherwise would be the best way of testing the effect of deep and shallow convection parameterisations on aerosol transport. Reducing entrainment can have a strong effect on the characteristics of parameterised convective clouds (see e.g. Labbouz et al., 2018). -L 331-332: Again, this statement is not true, according to figure 8. More description and analysis should be provided here: noshallowconv leads to an increase in both extinction and number concentrations in all regions, however turning off convective entrainment leads also to an increase in number concentration, but to either no changes or even a decrease in extinction. - L 331-332: Fig. 9 is barely described. Is it really needed in the paper? I would suggest either to remove it, or to go much further in the analysis. What can be gained from it? How can it help in understanding how changing convection affects aerosol vertical distributions?

Figure 8: as comparison between the absolute values of extinction in the different regions is not the main focus here, but rather the effect of changing model configurations, you may consider adapting the scale so that changes in extinction are more visible.

l337: that means no precipitation from warm clouds, hence possibly an overall reduction of wet scavenging.

Figure 11: again, why looking at cloud properties if not to go further in the analysis? The study focusses on aerosols, so I think discussing cloud properties is interesting only if they help better understand aerosol response (or lack of response). Otherwise, figure 11 could be deleted.

Section 4.3.5: why is the effect on extinction so small that it cannot be seen on the figure? You should discuss this result a bit more and try to explain it, especially as it is

different from Peers et al., 2016, as you mentioned in your conclusion.

L 367: showing observed extinction profiles on all of your figures would help assessing that more directly. This could be done by showing the markers, or indeed the standard deviation of the CALIOP profiles (based on the monthly-averaged, unless the comparison technique is changed following my suggestion of spatio-temporal colocation).

l395: Correct or clarify as it is almost impossible to see in most of the figures and it seems to be the opposite in the Peruvian BL.

l411-412: Modifying or turning off convective transport only (for shallow convection, deep convection, and both) would be an interesting sensitivity experiment.

L 481-482: some perspective could be added to actually give such a guidance. What should be done to improve the model? What are the next steps?

Technical corrections

The title could be improved, for instance changing it to "What are the processes controlling aerosol vertical distribution on Marine Stratocumulus region? A sensitivity study…." or to "Processes controlling the aerosol vertical distribution in five subtropical marine stratocumulus . . .". These are only suggestion, and I let the author decide whether they want to take any of them into account.

I think some commas should be deleted ( e.g. l 139 "We note here, that changes ..." the comma is confusing here)

l46-51: Could you try to rephrase this paragraph a little bit? The first part focusses a bit too much on everything being "important". Also, no so clear what is "its". Try to focuss on the main message here and rephrase to convey it in a simpler way.

L73: CALIOP: write what is stands for. L 77: a lidar is made of a laser and detector. You already said that CALIOP is a lidar, so I suggest deleting "using a lidar and detector"

l 84: remove further ; you could ,also remove the reference at the end of the sentence

(or replace "cf." by "following" )

l148: add "study" (after "sensitivity")

l149: replace "can not" by "cannot" (here and also in other occurences like l 289)

Figure 4: in the caption, replace the first "deviations" by "differences", and delete the second one : "Global distributions of differences in aerosol optical depth (left) and absorption aerosol optical depth (right) between the . . . " l 301, l 307, and other occurrences: avoid the use of "disabling" in this context, replace it by e.g. turning off l 302: I suggest replacing cut off by switched off or turned off (here and in all other occurences) figure 5 : the control simulation is fig 1 not fig 2 l 314: replace "disabled" by simply "no" (or "with wet deposition switched off") l 318: I suggest replacing by "in response to switching off wet deposition"

l414: "this convective scheme" ambiguous her (which one)?

References

Johnson, M. S., N. Meskhidze, and V. Praju Kiliyanpilakkil (2012), A global comparison of GEOS-Chempredicted and remotely-sensed mineral dust aerosol optical depth and extinction profiles, J. Adv.   Model.   Earth Syst., 4, M07001, doi:10.1029/2011MS000109.

Labbouz, L., Z. Kipling, P. Stier, and A. Protat (2018), How Well Can We Represent the Spectrum of Convective Clouds in a Climate Model?  Comparisons between Internal Parameterization Variables and Radar Observations. J. Atmos. Sci., 75, 1509–1524, https://doi.org/10.1175/JAS-D-17-0191.1

Peers, F., Bellouin, N., Waquet, F., Ducos, F., Goloub, P., Mollard, J., Myhre, G., Skeie, R. B., Takemura, T., Tanré, D., et al.  (2016), Comparison of aerosol optical properties above clouds between POLDER and AeroCom models over the South East Atlantic Ocean during the fire season, Geophys. Res. Lett., 43, 3991– 4000, doi:10.1002/2016GL068222.

Schutgens, N. A. J., Partridge, D. G., and Stier, P.: The importance of temporal collocation for the evaluation of aerosol models with observations, Atmos. Chem. Phys., 16, 1065–1079, https://doi.org/10.5194/acp-16-1065-2016, 2016.

Schutgens, N., Tsyro, S., Gryspeerdt, E., Goto, D., Weigum, N., Schulz, M., and Stier, P.: On the spatio-temporal representativeness of observations, Atmos. Chem. Phys., 17, 9761–9780, https://doi.org/10.5194/acp-17-9761-2017, 2017.

Wu, M., Liu, X., Zhang, L., Wu, C., Lu, Z., Ma, P.-L., et al. (2018). Impacts of aerosol dry deposition on black carbon spatial distributions and radiative effects in the Community Atmosphere Model CAM5. J. Adv. Model. Earth Syst., 10, 1150–1171. https://doi.org/10.1029/2017MS001219

---

## Referee Comment (RC2) · Anonymous Referee #1 · 10 Dec 2019

The manuscript presents a sensitivity study of the processes controlling the regional aerosol vertical distribution in the NorESM1-M model, with a particular focus on marine stratocumulus regimes and using satellite lidar retrievals from CALIOP as an observational reference. While the analysis draws significantly on previous studies such as Kipling et al. (2016) which carried out similar sensitivity tests in another model focussing on the global scale, the present manuscript adds a significant and welcome new element in bringing this approach together with vertically-resolved observations. This combination of model sensitivity referenced to observations is then a valuable

extension to the existing literature on aerosol vertical profiles, and I'm pleased to recommend it for publication in ACP subject to the following minor comments:

**Specific comments**

**p.2, line 25–26:** the Twomey and Albrecht effects are not the only proposed indirect effects or rapid adjustments contributing to ERFaci – there are several others relating to ice nucleation, glaciation and the invigoration or suppression of convection. Some of these remain quite speculative, but not necessarily any more so than the "cloud lifetime" interpretation of warm rain suppression.

**p.3, line 87:** why is a lower threshold required here rather than only the upper one? Wouldn't a CAD score lower than $-80$ be even more certain to be aerosol rather than cloud?

**p.4, line 101:** please specify the type of interpolation used (linear in height coordinates?)

**p.4, line 115:** please specify approximately how high "the lowest eight model levels" reaches, and the profile applied (equal mass per model level? uniformly in height or pressure coordinates?)

**p.4, line 122:** is the $r_{eff}$ dependence prognostic via a size-resolved cloud scheme, or is it diagnosed separately at each time step from the aerosol?

**p.5, line 140** : please explain briefly *why* the single-process approach is appropriate here, e.g. because many of the tests are not easily framed in a parametric way.

**p.6, lines 166–172:** this paragraph is a bit unclear. Do the terms "emission levels", "model emission levels" and "predefined emission levels" here all refer equivalently to the set of the lowest eight model levels (extending from the surface to

Interactive
comment
approximately 510 hPa)? In the last case, please specify the approximate height or pressure range spanned by the lowest three levels.

**p.6, lines 184–185:** it should be clarified that in-cloud scavenging refers to nucleation and impaction *by cloud droplets*, while below-cloud refers to *impaction by falling raindrops/precipitation*. It should probably be mentioned explicitly if either in-cloud scavenging by cloud ice particles or below-cloud scavenging by falling ice/snow/hail/graupel is or is not included in the model.

**p.7, lines 209–210:** 10 ms$^{-1}$ is already a very strong updraught velocity outside of deep convection, and 30 ms$^{-1}$ even more so. Given the focus here is on stratocumulus regimes, which are usually characterised by lower velocities, please check if these values are correct and if so consider the impact that this choice might have on the results. (They might be expected to produce large supersaturations and thus activate aerosols down to a smaller size than would occur with a more realistic stratocumulus vertical velocity.)

**p.8, lines 226–227:** the approach taken to checking significance against the variability in the data should be briefly mentioned here (it's very welcome that this is indeed considered as the results are presented).

**p.8, line 241:** again, please clarify the type of interpolation used.

**p.9, line 247:** what is meant by an "increase in magnitude in the boundary layer" here, where the text is talking about a single data set rather than comparing two? Does this mean "increasing with height away from the surface"?

**p.9, line 259:** the limited model resolution may still be important here: even if a layer or plume can be instantaneously represented at that resolution, it may be lost to diffusion too quickly.

**p.10, lines 285–286:** if this is the strongest response, it's surprising that it's not shown.

**p.10, line 303:** it's surprising that dry deposition has relatively little impact even in regions where dust and/or sea-salt are significant components. Do the authors have an explanation for this, given that dry deposition is usually a major sink process for these species? (Unlike the finer particles for which, as is stated, in-cloud wet deposition normally dominates.)

**p.10, lines 310–311:** again, what is meant by "decrease of aerosol extinction in the boundary layer" in the control simulation (not in something else relative to the control)? Does this mean a profile which decreases with height away from the surface? Please clarify.

**p.11, lines 315–316:** might a shift in size as well as composition be significant here?

**p.11, lines 340–344:** Figure 11 also seems to show a change in the cloud top height, which ought to be discussed.

**p.12, lines 358–360:** as mentioned above, increased model diffusion at limited resolution may play a role here.

**p.12, lines 371–372:** if the local maximum simply cannot be resolved at this vertical resolution it's unsurprising that none of the model configurations can reproduce it.

**p.13, line 396:** nucleation scavenging is efficient at removing large particles too (at least the soluble ones like coarse sea salt). Isn't it just that dry deposition and sedimentation are *also* efficient for these, where as they play little role for fine particles?

**p.14, line 413:** deep convection may still be allowed in the model, but does it actually play any role in the stratocumulus regimes that are the focus of this study?

**p.14, lines 429–434:** see also White et al. (2019), who show that the difference between microphysics schemes (and their autoconversion in particular) can be greater than the non-albedo aerosol indirect effects themselves; and West et al. (2014),

who demonstrate the importance of sub-grid vertical velocity variability in another model.

**p.14, lines 441–442:** "aerosol above clouds in climate models underestimate absorption" doesn't make sense. Please rephrase to clarify – it's not the aerosol that does the estimating.

**Figure 4:** do the boxes represent the regions referred to in the text? If so, please state this in the caption and label them. There's also a missing "of" in the caption (should be "Global distribution *of* deviations...").

**Figures 1, 5, 7:** it would be helpful if the boxes for the regions were also drawn on these figures, as on Figure 4, and the control included alongside each set for reference to avoid having to go back to Figure 1 on an earlier page to compare.

**Figures 3, 6, 8, 9, 10, 12:** There are a lot of lines with very similar colours on each of these. While there is a logic to using similar colours for each group of processes, this makes the plots harder to read as the lines on each plot are harder to distinguish. Since the groups are each plotted separately, using contrasting colours on each plot would make them more legible. If it's possible to reduce the number of lines further or adjust the scales to improve clarity that would also be welcome.

**Figures 9, 11:** more than half the vertical extent of these plots is unused – consider adjusting the vertical axis for the plots that don't go above the stratocumulus cloud top.

**Figures 9, 11, 12:** these plots are labelled with "Pressure (hPa)" on the vertical axis, but the same range (0–10) as the others using "Height (km)". Please check and ensure these are all labelled correctly and consistently.

**Technical corrections**

**p.1, line 12:** delete comma after "model levels".

**p.1, line 19:** delete comma after "heating".

**p.2, line 22:** "amount of liquid water content" ⟶ simply "liquid water content".

**p.2, line 29:** "that requires" ⟶ "which requires".

**p.3, line 80 and throughout:** "cf." is used repeatedly to introduce citations where it is probably not appropriate.

**References**

West, R. E. L., Stier, P., Jones, A., Johnson, C. E., Mann, G. W., Bellouin, N., Partridge, D. G., and Kipling, Z.: The importance of vertical velocity variability for estimates of the indirect aerosol effects, Atmos. Chem. Phys., 14, 6369–6393, https://doi.org/10.5194/acp-14-6369-2014, 2014.

White, B., Gryspeerdt, E., Stier, P., Morrison, H., Thompson, G., and Kipling, Z.: Uncertainty from the choice of microphysics scheme in convection-permitting models significantly exceeds aerosol effects, Atmos. Chem. Phys., 17, 12145–12175, https://doi.org/10.5194/acp-17-12145-2017, 2017.

---

## Author Comment (AC1) · 4 Nov 2020

**Response review #1**

The manuscript presents a sensitivity study of the processes controlling the regional aerosol vertical distribution in the NorESM1-M model, with a particular focus on marine stratocumulus regimes and using satellite lidar retrievals from CALIOP as an observational reference. While the analysis draws significantly on previous studies such as Kipling et al. (2016) which carried out similar sensitivity tests in another model focusing on the global scale, the present manuscript adds a significant and welcome new element in bringing this approach together with vertically-resolved observations. This combination of model sensitivity referenced to observations is then a valuable extension to the existing literature on aerosol vertical profiles, and I'm pleased to recommend it for publication in ACP subject to the following minor comments:

We would like to thank reviewer #1 for his/her comments, which improved the manuscript. We address each of the comments in the following.

Specific comments

- p.2, line 25–26: the Twomey and Albrecht effects are not the only proposed indirect effects or rapid adjustments contributing to ERFaci – there are several others relating to ice nucleation, glaciation and the invigoration or suppression of convection. Some of these remain quite speculative, but not necessarily any more so than the "cloud lifetime" interpretation of warm rain suppression.
  Thank you for your comment. We rephrased the sentence.

- p.3, line 87: why is a lower threshold required here rather than only the upper one? Wouldn't a CAD score lower than -80 be even more certain to be aerosol rather than cloud?
  Yes, no lower threshold is required and we used here only values below -80. Apologies, this was a mistake in the manuscript. We have tested different CAD scores along the project and failed to update the used score in the manuscript.

- p.4, line 101: please specify the type of interpolation used (linear in height coordinates?)
  Yes, we used linear interpolation in height coordinates and added this information in the manuscript.

- p.4, line 115: please specify approximately how high "the lowest eight model levels" reaches, and the profile applied (equal mass per model level? uniformly in height or pressure coordinates?)
  The lowest eight model levels reach up to approximately 5,5 km on average. We have added this information to the manuscript. The default IPCC emissions are distributed following the recommendations by Dentener et al. (2006), see Seland et al. (2008).

- p.4, line 122: is the $r_{eff}$ dependence prognostic via a size-resolved cloud scheme, or is it diagnosed separately at each time step from the aerosol?
  The $_{reff}$ dependence is prognostic and depends on the cloud droplet number concentration. We rephrased.

- p.5, line 140: please explain briefly why the single-process approach is appropriate here, e.g. because many of the tests are not easily framed in a parametric way.
  To simply identify processes which are controlling the vertical aerosol distribution, a simple on/off approach is more feasible. Once important processes are identified a parametric way would help to improve certain processes by testing parameter ranges. The aim of our study is to identify processes and emphasize their importance. We clarified this in the manuscript.

- p.6, lines 166–172: this paragraph is a bit unclear. Do the terms "emission levels", "model emission levels" and "predefined emission levels" here all refer equivalently to the set of the lowest eight model levels (extending from the surface to approximately 510 hPa)? In the last case, please specify the approximate height or pressure range spanned by the lowest three levels.
  We rephrased to clarify and added the pressure range for the lowest three model levels.

- p.6, lines 184-185: it should be clarified that in-cloud scavenging refers to nucleation and impaction by cloud droplets, while below-cloud refers to impaction by falling raindrops/precipitation. It should probably be mentioned explicitly if either in-cloud scavenging by cloud ice particles or below-cloud scavenging by fallingice/snow/hail/graupel is or is not included in the model.
  Thank you. We clarified in-cloud and below-cloud scavenging according to your suggestion. Scavenging in NorESM1 is only included for precipitation of liquid water, see Seland et al. [2008].

- p.7, lines 209-210: 10 ms-1 is already a very strong updraught velocity outside of deep convection, and 30 ms-1 even more so. Given the focus here is on stratocumulus regimes, which are usually characterised by lower velocities, please check if these values are correct and if so consider the impact that this choice might have on the results. (They might be expected to produce large supersaturations and thus activate aerosols down to a smaller size than would occur with a more realistic stratocumulus vertical velocity.)
  Yes, indeed, 30 ms$^{-1}$ is an extreme scenario and not a realistic case for the chosen stratocumulus regimes. We have clarified this in the manuscript. We chose this high velocity, since lower values within a more moderate range did not lead to a significant change in the simulated profiles.

- p.8, lines 226-227: the approach taken to checking significance against the variability in the data should be briefly mentioned here (it's very welcome that this is indeed considered as the results are presented).
  We have moved the explanation of our approach to test the statistical significance of sensitivity changes to Section 3.3.

- p.8, line 241: again, please clarify the type of interpolation used.
  We specified the type of interpolation used.

- p.9, line 247: what is meant by an "increase in magnitude in the boundary layer" here, where the text is talking about a single data set rather than comparing two? Does this mean "increasing with height away from the surface"?
  Yes, we meant an increase in magnitude with height. We clarified this.

- p.9, line 259: the limited model resolution may still be important here: even if a layer or plume can be instantaneously represented at that resolution, it may be lost to diffusion too quickly.
  Thank you. We have added your comment to the manuscript.

- p.10, lines 285-286: if this is the strongest response, it's surprising that it's not shown. The experiment with increased sizes of primary emitted particles shows a strong response in the Canarian region compared to the other regions, but it has not the strongest response in the Canarian region compared to other experiments. We rephrased the sentence and also show the results of this experiment in Figure 3.

- p.10, line 303: it's surprising that dry deposition has relatively little impact even in regions where dust and/or sea-salt are significant components. Do the authors have an explanation for this, given that dry deposition is usually a major sink process for these species? (Unlike the finer particles for which, as is stated, in-cloud wet deposition normally dominates.) In the experiment with dry deposition turned off, the model compensates the missing dry removal with an increased wet deposition. However, the opposite is not true for the experiment with wet deposition turned off, since dry removal is more efficient for larger particles. We have added this explanation to the manuscript.

- p.10, lines 310–311: again, what is meant by "decrease of aerosol extinction in the boundary layer" in the control simulation (not in something else relative to the control)? Does this mean a profile which decreases with height away from the surface? Please clarify
  Yes, we meant a decrease in aerosol extinction with height away from the surface. We rephrased the sentence.

- p.11, lines 315–316: might a shift in size as well as composition be significant here?
  Yes. We rephrased.

- p.11, lines 340–344: Figure 11 also seems to show a change in the cloud top height,which ought to be discussed.
  Yes, you are right. We removed Figures 9 and 11, following the recommendations by reviewer #2. We decided that it is easier to follow the manuscript by focusing on the vertical aerosol extinction distribution, rather than to elaborate on cloud properties.

- p.12, lines 358–360: as mentioned above, increased model diffusion at limited resolution may play a role here.
  Thank you. We included your comment in the manuscript.

- p.12, lines 371–372: if the local maximum simply cannot be resolved at this vertical resolution it's unsurprising that none of the model configurations can reproduce it.
  Yes, you are right. We removed this part of the sentence.

- p.13, line 396: nucleation scavenging is efficient at removing large particles too (at least the soluble ones like coarse sea salt). Isn't it just that dry deposition and sedimentation are also efficient for these, where as they play little role for fine particles?
  Yes, you are right. We rephrased the sentences.

- p.14, line 413: deep convection may still be allowed in the model, but does it actually play any role in the stratocumulus regimes that are the focus of this study?
  In general, deep convection does not play a role in the stratocumulus regimes. However, one should be aware that switching off shallow convection still allows deep convection and transport of aerosols.

- p.14, lines 429–434: see also White et al. (2019), who show that the difference between microphysics schemes (and their autoconversion in particular) can be greater than the non-albedo aerosol indirect effects themselves; and West et al. (2014), who demonstrate the importance of sub-grid vertical velocity variability in another model.
  Thank you for the references. We included them in the manuscript.

- p.14, lines 441–442: "aerosol above clouds in climate models underestimate absorption" doesn't make sense. Please rephrase to clarify – it's not the aerosol that does the estimating.
  Thank you. We rephrased the sentence.

- Figure 4: do the boxes represent the regions referred to in the text? If so, please state this in the caption and label them. There's also a missing "of" in the caption(should be "Global distribution of deviations. . . ").
  Yes, the boxes represent the regions. We added labels and adjusted the figure caption.

- Figures 1, 5, 7: it would be helpful if the boxes for the regions were also drawn on these figures, as on Figure 4, and the control included alongside

each set for reference to avoid having to go back to Figure 1 on an earlier page to compare.

We added boxes indicating the regions and also included the control simulation in Figures 5 and 7.

- Figures 3, 6, 8, 9, 10, 12: There are a lot of lines with very similar colours on each of these. While there is a logic to using similar colours for each group of processes, this makes the plots harder to read as the lines on each plot are harder to distinguish. Since the groups are each plotted separately, using contrasting colours on each plot would make them more legible. If it's possible to reduce the number of lines further or adjust the scales to improve clarity that would also be welcome.

We intended to have similar colors within one experiment category. We have now chosen more contrasting colors for the different profiles.

- Figures 9, 11: more than half the vertical extent of these plots is unused - consider adjusting the vertical axis for the plots that don't go above the stratocumulus cloud top.

We have chosen 10 km as an upper limit on the vertical axis following Koffi et al. (2016). But since the study by Koffi et al. (2016) has a different study domain and we focus only on marine stratocumulus regions, we have adjusted now the vertical axis.

- Figures 9, 11, 12: these plots are labelled with "Pressure (hPa)" on the vertical axis,but the same range (0–10) as the others using "Height (km)". Please check and ensure these are all labelled correctly and consistently.

We removed Figures 9 and 11, following the recommendation by reviewer #2. We corrected the label on Figure 12 (now Figure 10).

Technical corrections

- p.1, line 12: delete comma after "model levels".
Done.

- p.1, line 19: delete comma after "heating".
Done.

- p.2, line 22: "amount of liquid water content"→ simply "liquid water content".
Done.

- p.2, line 29: "that requires"→ "which requires.
Done.

- p.3, line 80 and throughout: "cf." is used repeatedly to introduce citations where it is probably not appropriate.
Done.

- References

West, R. E. L., Stier, P., Jones, A., Johnson, C. E., Mann, G. W., Bellouin, N.,Partridge, D. G., and Kipling, Z.: The importance of vertical velocity variability for estimates of the indirect aerosol effects, Atmos. Chem. Phys., 14, 369–6393,https://doi.org/10.5194/acp-14-6369-2014, 2014.
White, B., Gryspeerdt, E., Stier, P., Morrison, H., Thompson, G., and Kipling, Z.:Uncertainty from the choice of microphysics scheme in convection-permitting models significantly exceeds aerosol effects, Atmos. Chem. Phys., 17, 12145–12175,https://doi.org/10.5194/acp-17-12145-2017, 2017.Interactive comment on Atmos. Chem. Phys. Discuss., h

Thank you for the references. We included them in the manuscript.

**References**

Ø. Seland, T. Iversen, A. Kirkevåg, and T. Storelvmo. Aerosol-climate interactions in the cam-oslo atmospheric gcm and investigation of associated basic shortcomings. *Tellus A: Dynamic Meteorology and Oceanography*, 60(3):459–491, 2008. doi: 10.1111/j.1600-0870.2007.00318.x.

---

## Author Comment (AC2) · 5 Nov 2020

**Referee #2**

General Comments
This article provides an interesting sensitivity study exploring the effect of changes in model parameters and aerosol emissions on aerosol composition and vertical distribution of extinction and number concentration, focussing on the marine stratocumulus regions. It analyses separately the impact of changing parameters one by one in the simulations, and concludes on the relative importance of the processes considered,showing that although some of them like the wet scavenging have a strong impact,none is able to reproduce the CALIOP observations.
I think the analysis could be deepen and the interpretation of the results would gain in being extended. Even if the full chain of processes is very complex to analyses in a climate model, and without providing a full pathway analysis that would need extensive additional work, I think more insight could be gained by crossing the results and trying to interpret them (especially when they are surprising or when there are regional differences).More direct comparisons with the observations could be provided to asses the effect of parameter changes, and spatial and temporal colocation could increase the robustness of the comparison (although they might not be straightforward to implement). More highlights could be put on answering the question: Could the model possibly represent better the observations if the relevant parameters where adjusted ? Would this set of parameters be realistic? Or are there fundamental discrepancies that cannot be resolved by parameter changes?I think more simulations could be performed to either better distinguish between processes (convection parameterisation vs. aerosol transport by convection for instance)or investigate other key properties of the model, like its vertical resolution which could be essential in representing the low-level aerosol distribution. More details could also be provided on the model setup, on the characteristics of the parameterisations, and the choices made for the sensitivity study.Please refer to my specific comments here-after for more details.The paper is well written overall (some English editing is needed here and there, cf.my technical comments) and is organised in a straightforward way. Although I have numerous comments providing ways for clarifications and improvements, I believe this paper is a good contribution to the literature and I am sure its revised form will be publishable in Atmospheric Chemistry and Physics.

We would like to thank reviewer #2 for all the comments and suggestions, which improved the manuscript. We have expanded on interpretation of some of the results that are unexpected (like the lack of sensitivity to dry deposition). With regard to the comparison with observations, the combination of model resolution and available observations does not allow for directly colocated point-by-point comparisons, and we choose to look at a regional and annual mean scale; we have however added comment on this in the manuscript. We have also added discussion on the overarching questions of whether it is feasible to adjust model parameters in a realistic way to create better agreement with observations. More details on the model setup, and parameterisations has been added as well. We

answer to each of the individual comments in the following.

Specific comments

- L11-13: is that really resolution that matters here or rather proper coloca-tion? Similarly, not sure about the relevance of interpolation (cf comments hereafter).
  A proper temporal and spatial colocation as you suggested later is not possible since the model output are monthly means. The improved agree-ment with the interpolated observations suggests therefore that the model resolution is important here.

- L 80: "... underestimation of aerosols near the surface" any reference supporting this statement?
  We rephrased the sentence.

- L99: To be fully consistent, model data should be also extracted along CALIPSO over-passes, at the times of the overpasses, before being aver-aged. Although daily mean works rather well in areas where there is no strong diurnal cycle in aerosols, proper spatial and temporal colocation (of the model data onto CALIOP measurements) reduces errors (cf. e.g. Schutgens et al., 2017). It may not be easily doable to extract profiles along CALIPSO track from the model, but discussions of sampling errors could be included.
  Thank you. The model output are monthly means, so that we cannot ex-tract model output along the CALIPSO track at the overpass times. We added the reference in Section 3.3. and point to possible sampling errors.

- L101-102: Is it really interpolation that is used here? As the CALIPSO data are on a finer vertical grid than the model, it would be better to average all the CALIOP data points located inside one model gridbox than to interpolate between two CALIOP levels to get the value at the central point of the gridbox.
  At the beginning of the project, we started with averaging the vertical levels of CALIOP as you suggested, but have then decided to choose linear interpolation instead. By averaging, parts of the original shape of the observed CALIOP profile would be lost. Linear interpolation seems to be the better method for containing the original shape but to still guarantee a more fair comparison for the model, which has a much lower resolution. We have added this information to the manuscript.

- L 110-111: could you please show the location of the vertical model levels at least in one of your plots (e.g. adding markers figure 1) or / and give the spacing between levels in the low to mid troposphere?
  Following previous publications and for a better visibility, we do not show the model levels in the figures, but instead state the pressure range for the model levels in the text.

- L 115: "the lowest eight levels" corresponding to what altitude (on average)?
  The lowest eight model levels are corresponding to a pressure range from the surface to approximately 510 hPa on average. We have added this information to the manuscript.

- L 121: be more specific: the cloud albedo and cloud lifetime effects are not directly parameterised, but the microphysics parameterisation takes aerosol into accounts and hence aims to represent them.
  Yes, you are right. We rephrased the sentence.

- L 125: why is there a maximum precipitation rate? It seem odd if you do not specify here (as line 205) "before the autoconversion is switched off".
  We apologise, our sentence was misleading. The critical precipitation rate is not triggering autoconversion. If the critical precipitation rate is reached, the collector drops are assumed to influence the drop size and thereby autoconversion. We rephrased the sentence.

- L 127: what means "production-tagged"?
  The aerosol life cycle scheme in NorESM is production-tagged, i.e. the different emitted particles will be "tagged" with a production mechanism, such as e.g. nucleation. We rephrased the sentence.

- L 129: "for convective clouds an in-plume approach is used i.e. the convective cloud cover is calculated explicitly": explain a bit more. What do you mean by "in-plume approach" how is calculated the convective cloud cover? How is it then passed to the large-scale? As convective clouds are parameterised, their cloud cover is surely not fully explicit. As there is no aerosol in the Zhang and McFarlane (1995) scheme, could you be more precise and, if they have been added in a more recent version, cite the relevant literature?
  Yes, you are right. The convective cloud cover is not fully explicit. There is a distinction between an in-plume and an operator-split approach. An operator split approach means that processes are acting sequentially, while an in-plume approach allows processes to act simultaneously. In NorESM1, aerosols can be vertically transported, mixed between updrafts and downdrafts and removed directly with wet scavenging [Kirkevåg et al., 2013].

- L 134: be more specific on the characteristics of the run, and/or give reference for AMIP setup.
  An AMIP setup uses prescribed sea surface temperatures and sea ice from 1980 to present-day. We rephrased the sentence.

- L 159-165: Justify the choice of this emission dataset. Are they more realistic for the simulation period? Why not using realistic monthly emissions for the period of simulations as a control? And then either a different dataset, or a multiplicative factor on emissions for the sensitivity experiment?

The AMIP setup for our simulations with prescribed sea ice and sea surface temperatures does not allow transient aerosol emissions, i.e. synchronous with the actual year. To at least study the effect of more recent emissions, we included the emission dataset with emissions available until 2010. We have added this motivation to the manuscript.

- L 183: are aerosols also liberated by evaporation of cloud droplets and raindrops? If yes, you could mention it in paragraph 3.1.
  Yes, aerosols are liberated by evaporation of cloud droplets (see Kirkevåg et al. [2013]). We added this information to the manuscript.

- L 189: the original convection scheme should be described a bit more (here, or maybe rather section 3.1). Do you mean only deep convection here? What mean the full mixing of aerosols? the aerosol population is the same in updraughts and downdraughts? How about the impact of lateral entrainment then? By "the original scheme" do you mean Zhang and McFarlane (1995) which has no aerosol at all?
  In the experiment Aero2000_convmix shear-generated turbulence fully mixes constituents between the up- and downdrafts of convective clouds (see Seland et al. [2008]). The mass fluxes are thereby based on Zhang and McFarlane [1995].

- L190-196: - What happens in the model when shallow convection is turned off? Is it picked-up by the deep convection scheme? Or by the large-scale as it tends to be when all convection parameterisation if turned off? In any case, turning off shallow convection will not prevent the vertical transport needed to balance surface SW heating and atmospheric LW cooling. - Then, why not turning off only aerosol transport from convection parameterisation (looking at both shallow and deep convection separately)? That would give much clearer results on what is done by the parametrisation in term of aerosols, without having any direct impact on the dynamics, clouds, etc.
  If shallow convection is turned off, the deep convection scheme takes over. We agree, that only switching off aerosol transport would be useful and was originally planned following Kipling et al. (2016), but this was unfortunately not possible in the model.

- L200: what schemes are used? More description of the original scheme is needed (here or in section 3.1) before discussing its perturbations.
  The autoconversion scheme is based on Tripoli and Cotton [1979] and modified for the model by Rasch and Kristjánsson [1998]. We clarified this in the manuscript in section 3.1.

- L 204-206: You could include the equations for autoconversion (and possibly accretion). This threshold on the radius has been introduced in models historically, partly to compensate for the lack of below-cloud evaporation, but there should not be any threshold as the processes are continuous. The threshold in precipitation is even more arbitrary as cloud droplet should

continue to form raindrops no matter how much precipitation there is already (although accretion will then become much more significant than autoconversion, meaning in practice autoconversion might be of little or no effect). Unless the way the equations are written makes it unphysical, I would suggest trying to remove the two thresholds.
As stated earlier, the autoconversion scheme is based on Tripoli and Cotton [1979] and modified by Rasch and Kristjánsson [1998]. We added these references, rather than introducing an equation. Removing the thresholds was not possible and also a simulation with unrealistic high thresholds did not work.

- L 208: what is the "characteristic subgrid vertical velocity"? What will be the effect of changing it? Explain so that the reader can understand what the chosen values mean.
  Rather than taking a mean for a grid box, a subrgid vertical velocity is defined to represent the variability within one model grid box. The vertical velocity is needed for the activation of clouds droplets. The subgrid vertical velocity is defined as $w' = \frac{K_d}{/} l_c$, see Morrison and Gettelman [2008]. We clarified this in the manuscript.

- L 209: "high variability" in what sense?
  We meant the standard deviation range of the control simulation. We have run several experiments with different values for the subgrid vertical velocity. Since chosing realisting values didn't lead to a strong response in the model, we chose to illustrate the influence of vertical velocity with the extreme value of 30ms.

- L 220: is the monthly output obtained from online averaging over the month?
  Yes.

- L 222-225: following my previous comment, is that also true for temporal sampling? Schutgens et al. (2016, 2017) suggest the opposite.
  As stated earlier, model output is available only as monthly means, so that temporal sampling at the CALIOP overpass times is not possible.

- L241: again, is it really an interpolation? Averaging would be better.
  See comment above. We chose interpolation rather than averaging to allow a more fair comparison between the model and the observations.

- L 249: what is the average BL height in these regions The average height is approximately 850 hPa.

- L 256: Indeed model resolution is too coarse (and probably also in the free troposphere up to 5.5 km). From figure 2, I guess AOD is also underestimated by the model? An interesting additional sensitivity experiment could be to refine the vertical grid in the BL and up to about 5.5 km. The model version is only available for 30 vertical levels and it is not possible to increase the number in levels.

- Section 4.3: why not include CALIOP extinction profiles in the figures? This would be useful to compare not only sensitivity experiments with the control but also with the observations and see when they perform better than the control. Indeed, one big question is whether or not changes in model configuration can lead to results closer to the observations, so a more direct comparison is needed in the figure and in the analysis.
  We show the CALIOP profile now also in Figures 2, 6, 8, 9 and 10.

- L 283: "Hence, by changing the size of ..." rephrase to make it clearer, e.g. "Hence, changing the size of emitted particles also leads to changes in emitted aerosol numbers"
  Thanks. We rephrased the sentence.

- l 289-290: "As a consequence...." I do not understand. Something is not right in the way this sentence in constructed. Please clarify / rephrase.
  We rephrased the sentence.

- L 303: The differences from turning off dry deposition are actually almost non-existent, indicating that the dry deposition plays very little role (if any) in your simulations. Although dry deposition will affect mostly the biggest aerosols, I am a bit surprised that the impact is so small. How big is the impact on the total aerosol burden? Using CAMS, Wu et al. (2018) show significant impact of the dry deposition scheme on BC burden (cf.for instance their figure4), and I suppose this could also be the case for dust (Johnson et al., 2012 ). Could you discuss that a bit? Do you think the dry deposition could be underestimated in your control simulation?
  The model compensates the lack of dry removal by wet deposition. Wet deposition is increased in the experiment with dry deposition turned off. The opposite is not the case for the experiment with switched off wet deposition, which makes sense, since dry deposition is more efficient for larger particles.

- L 312: again, it would be helpful to plot the observed extinction profiles on the same figure.
  We added the CALIOP profiles to Figures 3, 6, 8, 9 and 10.

- L319-320: can you explain and justify this statement? How do you know the composition changes affects extinction more than the number concentration?

- L320: again, I am surprised by the total lack of sensitivity to dry deposition.
  See comment above. Wet deposition increases when dry deposition is turned off.

- Section 4.3.3: more careful description and analysis is needed: -
  L326-327: No, there is no decrease in aerosol number above the BL according to fig 8. In all regions and at all heights, there is an increase in both

extinction and number concentration. Can you interpret that? It might be related to other changes in the simulation without shallow convection; turning off only aerosol transport by shallow convection would make the interpretation easier.

Yes, you are right. We corrected the sentence.

- L 329: do you mean you switched off entrainment completely in shallow and deep convective clouds (no lateral or below cloud entrainment of aerosol, momentum, environmental air, etc)? Turning off only the transport of aerosols by convection, but keeping entrainment unchanged otherwise would be the best way of testing the effect of deep and shallow convection parameterisations on aerosol transport. Reducing entrainment can have a strong effect on the characteristics of parameterised convective clouds (see e.g. Labbouz et al., 2018).

  Yes, entrainment was switched off completely. We agree, it would be better to switch off only convective transport, but as stated earlier, this is not possible. Thank you for the reference. We included it in the manuscript.

- L 331-332: Again, this statement is not true, according to figure 8. More description and analysis should be provided here: noshallowconv leads to an increase in both extinction and number concentrations in all regions, however turning off convective entrainment leads also to an increase in number concentration, but to either no changes or even a decrease in extinction.

  We corrected the sentence.

- L331-332: Fig.9 is barely described. Is it really needed in the paper? I would suggest either to remove it, or to go much further in the analysis. What can be gained from it? How can it help in understanding how changing convection affects aerosol vertical distributions?

  You are right. The figure is not needed in the paper and we removed it.

- Figure 8: as comparison between the absolute values of extinction in the different regions is not the main focus here, but rather the effect of changing model configurations, you may consider adapting the scale so that changes in extinction are more visible.

  We adjusted the scale to make changes in the vertical distribution more visible.

- L337: that means no precipitation from warm clouds, hence possibly an overall reduction of wet scavenging.

  Yes, the wet scavenging in this simulation is reduced. We state this now in the manuscript.

- Figure 11: again, why looking at cloud properties if not to go further in the analysis? The study focuses on aerosols, so I think discussing cloud properties is interesting only if they help better understand aerosol response (or lack of response). Otherwise, figure 11 could be deleted. Yes.

You are right. We decided to remove Figures 9 and 11, since they do not help in understanding the aerosol distribution.

- Section 4.3.5: why is the effect on extinction so small that it cannot be seen on the figure? You should discuss this result a bit more and try to explain it, especially as it is different from Peers et al., 2016, as you mentioned in your conclusion.
  We clarified this in the manuscript.

- L 367: showing observed extinction profiles on all of your figures would help assessing that more directly. This could be done by showing the markers, or indeed the standard deviation of the CALIOP profiles (based on the monthly-averaged, unless the comparison technique is changed following my suggestion of spatio-temporal colocation).
  We added CALIOP profiles to Figures 3, 6, 8, 9 and 10.

- L395: Correct or clarify as it is almost impossible to see in most of the figures and it seems to be the opposite in the Peruvian BL.
  Yes, you are right, there is a small increase in the Peruvian BL. We corrected the statement.

- L411-412: Modifying or turning off convective transport only (for shallow convection, convection, and both) would be an interesting sensitivity experiment. Thank you. We agree and had also the idea to switch off only the transport when we started with the project, but there seems to be no way in the model to switch off only the convective transport.

- L 481-482: some perspective could be added to actually give such a guidance. What should be done to improve the model? What are the next steps?
  We have added some perspectives to the manuscript.

  Technical corrections.

- The title could be improved, for instance changing it to "What are the processes con-trolling aerosol vertical distribution on Marine Stratocumulus region? A sensitivity study...." or to "Processes controlling the aerosol vertical distribution in five subtropical marine stratocumulus...". These are only suggestion, and I let the author decide whether they want to take any of them into account.
  Thanks for the suggestions. We changed the title.

- I think some commas should be deleted ( e.g. L 139 "We note here, that changes ..."the comma is confusing here)
  We corrected the punctuation of commas in the manuscript.

- L46-51: Could you try to rephrase this paragraph a little bit? The first part focuses a bit too much on everything being "important". Also, no so

clear what is "its". Try to focus on the main message here and rephrase to convey it in a simpler way.
We rephrased the sentence.

- L73: CALIOP: write what is stands for.
  We already wrote in L 67 the abbreviation.

- L 77: a lidar is made of a laser and detector. You already said that CALIOP is a lidar, so I suggest deleting "using a lidar and detector".
  Done.

- L 84: remove further ; you could ,also remove the reference at the end of the sentence (or replace "cf." by "following" )
  Done. We replaced cf. with following.

- L148: add "study" (after "sensitivity")
  Done.

- L149: replace "can not" by "cannot" (here and also in other occurences like L 289)
  We changed it throughout the manuscript.

- Figure 4: in the caption, replace the first "deviations" by "differences", and delete the second one : "Global distributions of differences in aerosol optical depth (left) and absorption aerosol optical depth (right) between the..." Done.

- L 301, L 307, and other occurrences: avoid the use of "disabling" in this context, replace it by e.g. turning off Thank you. We replaced it throughout the manuscript.

- L 302: I suggest replacing cut off by switched off or turned off (here and in all other occurences)
  Thanks. We replaced cut off throughout the manuscript.

- Figure 5 : the control simulation is fig 1 not fig 2
  We have added the control simulation to Figure 5 and 7 and removed the cross-reference.

- L 314: replace "disabled" by simply "no" (or "with wet deposition switched off")
  Done.

- L 318: I suggest replacing by "in response to switching off wet deposition"
  Done.

- L 414: "this convective scheme" ambiguous her (which one)?
  Thanks. We meant the shallow convective scheme here and specified it now in the text.

- References

Johnson, M. S., N. Meskhidze, and V. Praju Kiliyanpilakkil (2012), A global comparison of GEOS-Chem predicted and remotely-sensed mineral dust aerosol op-tical depth and extinction profiles, J. Adv.Model.Earth Syst., 4, M07001,doi:10.1029/2011MS000109.

Labbouz, L., Z. Kipling, P. Stier, and A. Protat (2018), How Well Can We Represent the Spectrum of Convective Clouds in a Climate Model? Comparisons between Internal Parameterization Variables and Radar Observations. J. Atmos. Sci., 75, 1509–1524,https://doi.org/10.1175/JAS-D-17-0191.1

Peers, F., Bellouin, N., Waquet, F., Ducos, F., Goloub, P., Mollard, J., Myhre, G.,Skeie, R. B., Takemura, T., Tanré, D., et al. (2016), Comparison of aerosol optical properties above clouds between POLDER and AeroCom models over the SouthEast Atlantic Ocean during the fire season, Geophys. Res. Lett., 43, 3991– 4000,doi:10.1002/2016GL068222.

Schutgens, N. A. J., Partridge, D. G., and Stier, P.: The importance of temporal collocation for the evaluation of aerosol models with observations, Atmos. Chem. Phys.,16, 1065–1079, https://doi.org/10.5194/acp-16-1065-2016, 2016.

Schutgens, N., Tsyro, S., Gryspeerdt, E., Goto, D., Weigum, N., Schulz, M., and Stier,P.: On the spatio-temporal representativeness of observations, Atmos. Chem. Phys.,17, 9761–9780, https://doi.org/10.5194/acp-17-9761-2017, 2017.Wu, M., Liu, X., Zhang, L., Wu, C., Lu, Z., Ma, P.-L., et al. (2018). Impacts of aerosol dry deposition on black carbon spatial distributions and radiative effects in the Community Atmosphere Model CAM5. J. Adv. Model. Earth Syst., 10, 1150–1171.https://doi.org/10.1029/2017MS001219

Thank you for the references. We included them in the manuscript.

**References**

A. Kirkevåg, T. Iversen, Ø. Seland, C. Hoose, J. E. Kristjánsson, H. Struthers, A. M. L. Ekman, S. Ghan, J. Griesfeller, E. D. Nilsson, and M. Schulz. Aerosol–climate interactions in the Norwegian Earth System Model – NorESM1-M. *Geosci. Model Dev.*, 6(1):207–244, 2013. ISSN 1991-9603. doi: 10.5194/gmd-6-207-2013.

Hugh Morrison and Andrew Gettelman. A new two-moment bulk stratiform cloud microphysics scheme in the community atmosphere model, version 3 (cam3). part i: Description and numerical tests. *Journal of Climate*, 21(15):3642–3659, 2008. doi: 10.1175/2008JCLI2105.1. URL https://doi.org/10.1175/2008JCLI2105.1.

P. J. Rasch and J. E. Kristjánsson. A Comparison of the CCM3 Model Climate Using Diagnosed and Predicted Condensate Parameterizations. *Journal of Climate*, 11(7):1587–1614, 07 1998. ISSN 0894-8755. doi: 10.1175/1520-0442(1998)011¡1587:ACOTCM¿2.0.CO;2. URL `https://doi.org/10.1175/1520-0442(1998)011<1587:ACOTCM>2.0.CO;2`.

Ø. Seland, T. Iversen, A. Kirkevåg, and T. Storelvmo. Aerosol-climate interactions in the cam-oslo atmospheric gcm and investigation of associated basic shortcomings. *Tellus A: Dynamic Meteorology and Oceanography*, 60(3):459–491, 2008. doi: 10.1111/j.1600-0870.2007.00318.x.

Gregory J. Tripoli and William R. Cotton. A Numerical Investigation of Several Factors Contributing to the Observed Variable Intensity of Deep Convection over South Florida. *Journal of Applied Meteorology*, 19(9):1037–1063, 09 1979. ISSN 0021-8952. doi: 10.1175/1520-0450(1980)019¡1037:ANIOSF¿2.0.CO;2. URL `https://doi.org/10.1175/1520-0450(1980)019<1037:ANIOSF>2.0.CO;2`.

Guang J. Zhang and Norman A. McFarlane. Sensitivity of climate simulations to the parameterization of cumulus convection in the canadian climate centre general circulation model. *Atmosphere-ocean*, 33(3):407–446, 1995.